# The acidic domain of the endothelial membrane protein GPIHBP1 stabilizes lipoprotein lipase activity by preventing unfolding of its catalytic domain

Simon Mysling[1,2,3], Kristian Kølby Kristensen[1,2], Mikael Larsson[4], Anne P Beigneux[4], Henrik Gårdsvoll[1,2], Loren G Fong[4], André Bensadouen[5], Thomas JD Jørgensen[3], Stephen G Young[4,6], Michael Ploug[1,2]*

[1]Finsen Laboratory, Rigshospitalet, Copenhagen, Denmark; [2]Biotech Research and Innovation Centre, University of Copenhagen, Copenhagen, Denmark; [3]Department of Biochemistry and Molecular Biology, University of Southern Denmark, Odense, Denmark; [4]Department of Medicine, University of California, Los Angeles, Los Angeles, United States; [5]Division of Nutritional Science, Cornell University, Ithaca, United States; [6]Department of Human Genetics, University of California, Los Angeles, Los Angeles, United States

*For correspondence: m-ploug@finsenlab.dk

**Abstract** GPIHBP1 is a glycolipid-anchored membrane protein of capillary endothelial cells that binds lipoprotein lipase (LPL) within the interstitial space and shuttles it to the capillary lumen. The LPL•GPIHBP1 complex is responsible for margination of triglyceride-rich lipoproteins along capillaries and their lipolytic processing. The current work conceptualizes a model for the GPIHBP1•LPL interaction based on biophysical measurements with hydrogen-deuterium exchange/mass spectrometry, surface plasmon resonance, and zero-length cross-linking. According to this model, GPIHBP1 comprises two functionally distinct domains: (1) an intrinsically disordered acidic N-terminal domain; and (2) a folded C-terminal domain that tethers GPIHBP1 to the cell membrane by glycosylphosphatidylinositol. We demonstrate that these domains serve different roles in regulating the kinetics of LPL binding. Importantly, the acidic domain stabilizes LPL catalytic activity by mitigating the global unfolding of LPL's catalytic domain. This study provides a conceptual framework for understanding intravascular lipolysis and GPIHBP1 and LPL mutations causing familial chylomicronemia.

## Introduction

For nearly six decades, we have known that lipoprotein lipase (LPL), a triglyceride hydrolase secreted by myocytes and adipocytes, is essential for the lipolytic processing of triglyceride-rich lipoproteins (TRLs) in the bloodstream (*Korn, 1955*; *Young and Zechner, 2013*). LPL-mediated processing of TRLs occurs along the capillary lumen, and LPL is readily released from the surface of capillaries with polyanionic compounds such as heparin. LPL contains an N-terminal domain (NTD) for catalysis and a C-terminal domain (CTD) that is essential for lipid binding. LPL is active as a head-to-tail homo-dimer, and several lines of evidence suggest that the CTD of one LPL monomer delivers triglyceride substrates to the NTD of the partner monomer (*Kobayashi et al., 2002*). In the presence of certain LPL mutations (e.g., mutations in the catalytic triad or mutations that prevent LPL dimerization), the processing of TRLs is markedly impaired, leading to severe hypertriglyceridemia (familial chylomicro-nemia) (*Brahm and Hegele, 2015*).

**eLife digest** Fat is an important part of our diet. The intestines absorb fats and package them into particles called lipoproteins. After reaching the bloodstream, the fat molecules (lipids) in the lipoproteins are broken down by an enzyme called lipoprotein lipase (LPL), which is located along the surface of small blood vessels. This releases nutrients that can be used by vital tissues – mainly the heart, skeletal muscle, and adipose tissues. LPL is produced by muscle and adipose tissue, but it is quickly swept up by a protein called GPIHBP1 and shuttled to its site of action inside the blood vessels.

Mutations that alter the structure of LPL or GPIHBP1 can prevent the breakdown of lipids, resulting in high levels of lipids in the blood. This can lead to inflammation in the pancreas and also increases the risk of heart attacks and strokes. Many earlier studies have examined the properties of LPL, but our understanding of GPIHBP1 has been limited, mainly because it has been difficult to purify GPIHBP1 for analysis.

Using genetically altered insect cells, Mysling et al. were able to purify two different forms of GPIHBP1 – a full-length version and a shorter version that lacked a small section at the end of the molecule known as the acidic domain. This revealed that the opposite end of the molecule – called the carboxyl-terminal domain – is primarily responsible for binding LPL and anchoring it inside blood vessels. Once LPL is bound to GPIHBP1, the acidic domain of GPIHBP1 helps to stabilize LPL. If GPIHBP1's acidic domain is missing then LPL is more susceptible to losing its structure, rendering it incapable of breaking down the lipids in the blood.

Mysling et al. describe a new model for how LPL and GPIHBP1 interact that explains how specific mutations in the genes that encode these proteins interfere with the delivery of LPL to small blood vessels. In the future, this could help researchers to develop new strategies to treat people with high levels of lipids in their blood.

For decades, the mechanism by which LPL traversed endothelial cells to reach the luminal surface of capillaries—as well as LPL's binding site within capillaries—were enigmas (*Brown et al., 2015*). These uncertainties have been addressed by the discovery of a novel LPL binding protein of capillary endothelial cells, glycosylphosphatidylinositol-anchored high density lipoprotein–binding protein 1 (GPIHBP1). GPIHBP1 binds LPL within the interstitial spaces and transports LPL across endothelial cells to its site of action in the capillary lumen (*Beigneux et al., 2007*; *Davies et al., 2010*). In the absence of GPIHBP1, LPL does not reach the capillary lumen, and there is no margination of TRLs along the surface of capillaries (*Goulbourne et al., 2014*).

GPIHBP1 is a member of the LU (Ly6/uPAR) protein domain family, which includes CD59, SLURP1, C4.4A, TGF-β receptors, and uPAR. The hallmark of this protein family is a three-finger–fold with a defined disulfide pairing motif involving eight highly conserved cysteines (*Ploug, 2003*). Members of the LU domain family are widespread in the animal kingdom and occur as secreted proteins (e.g. SLURP1), integral membrane proteins (TGFβ receptors), and glycolipid-anchored proteins (e.g. CD59, GPIHBP1, C4.4A, uPAR). During evolution, the LU domain has evolved to serve many different purposes in mammals: GPIHBP1 transports LPL; CD59 inhibits complement activation; TGFβ receptors are involved in cytokine signaling; TEX101 regulates fertility; and uPAR focuses plasminogen activation on cell surfaces. Although detailed structure–function relationships have been elucidated for a few mammalian LU domain proteins (e.g. TGFβ receptors, CD59, uPAR), progress in understanding the biochemistry and biophysics of protein interactions for the majority of LU domain proteins, including GPIHBP1, has lagged behind, largely because of the need for highly purified recombinant protein preparations.

In our opinion, GPIHBP1 represents one of the more intriguing mammalian LU domain proteins from both a functional and a structural point of view. First, GPIHBP1 function is tightly linked to human disease as a variety of loss-of-function mutations in GPIHBP1 have been encountered in humans with familial chylomicronemia (*Beigneux et al., 2009*; *Buonuomo et al., 2015*; *Olivecrona et al., 2010*; *Plengpanich et al., 2014*; *Rios et al., 2012*; *Surendran et al., 2012*). The majority of these disease-causing mutations impair the folding of the LU domain, leading to

multimerized and dysfunctional GPIHBP1 molecules on the cell surface (*Beigneux et al., 2015*). In a comprehensive screening of GPIHBP1 mutants, one mutant (GPIHBP1$^{W89S}$) was found to have markedly impaired LPL binding despite being monomeric, sugge

sting that it retained the native folding of its LU domain (*Beigneux et al., 2011*; *Beigneux et al., 2015*). Interestingly, several mutations in the CTD of LPL (e.g. C418Y, first identified in a patient with chylomicronemia) have no effect on catalysis but abolish the ability of LPL to bind to GPIHBP1 (and thus prevent LPL transport across endothelial cells) (*Gin et al., 2012*; *Henderson et al., 1996*). Amino acid numbering throughout this article refers to the first residue in the mature protein.

Second, GPIHBP1 is unique from a structural perspective. Among the entire LU protein family, only very few proteins have N-terminal extensions (e.g. GPIHBP1 and the spermatid marker SP-10). In the case of GPIHBP1, the N-terminal extension is highly enriched in acidic residues, with 21 of 26 consecutive amino acids in human GPIHBP1 being aspartic acid or glutamic acid. Early studies in cell culture suggested that GPIHBP1's acidic domain is important for GPIHBP1•LPL interactions (*Gin et al., 2008*), but a detailed study of the functional impact of the acidic domain on LPL function has been lacking. One of the impediments to progress is that a large fraction of the GPIHBP1 produced by transfected mammalian cells is misfolded and in the form of multimers (*Beigneux et al., 2015*).

In the current study, we used highly purified proteins and a combination of hydrogen–deuterium exchange mass spectrometry (HDX-MS), surface plasmon resonance (SPR), zero-length cross-linking, and LPL activity assays to elucidate LPL•GPIHBP1 interactions both kinetically and dynamically. Our studies show: (1) that the kinetics of LPL•GPIHBP1 interactions are controlled by both the LU domain and the acidic domain of GPIHBP1; and (2) that the acidic domain is intrinsically disordered and is responsible for stabilizing the catalytic activity of LPL by inhibiting the inherent instability and subsequent unfolding of LPL's catalytic domain.

## Results

### Production and purification of recombinant soluble GPIHBP1

A secreted version of human GPIHBP1$^{1–131}$ was produced in *Drosophila* S2 cells as a fusion protein with uPAR domain III (*Figure 1—figure supplement 1*). However, this protein proved to be prone to an internal cleavage after Arg$^{38}$ during enterokinase-mediated removal of the uPAR tag. This unexpected cleavage event markedly reduced the yields of purified GPIHBP1$^{1–131}$. Alignments of multiple primate GPIHBP1 sequences revealed that Arg$^{38}$ is not conserved during evolution. In GPIHBP1 from *Nomascus leucogenys,* which is 94% identical to human GPIHBP1, the residue corresponding to Arg$^{38}$ is Gly$^{38}$. Based on these homology considerations, we therefore expressed and purified a modified protein in which Arg$^{38}$ was replaced with Gly; this construct yielded high levels of pure GPIHBP1$^{1–131/R38G}$ along with moderate amounts of a truncated GPIHBP1$^{34–131/R38G}$. That truncated protein was the result of an additional cleavage after Arg$^{33}$. Both GPIHBP1$^{1–131}$ and GPIHBP1$^{34–131}$ (lacking the acidic domain) were purified to homogeneity by cation-exchange chromatography (*Figure 1—figure supplement 2*). Importantly, both proteins were monomeric with no traces of aggregation, as judged by analytical size-exclusion chromatography (*Figure 1C*). This homogeneity is a noteworthy achievement because earlier studies had shown that GPIHBP1 is highly susceptible to multimerization (*Beigneux et al., 2015*). The anomalous partitioning of GPIHBP1$^{1–131}$ during size-exclusion chromatography is most likely a consequence of a large Stokes radius caused by the presence of an intrinsically disordered N-terminal peptide (see next section). Consistent with this assumption, GPIHBP1$^{34–131}$, which lacks the acidic domain, eluted with the expected hydrodynamic volume for a globular protein (*Figure 1C*).

### Molecular model for GPIHBP1

Homology considerations identify GPIHBP1 as a glycolipid-anchored protein with a single prototypical LU domain (*Figure 1A*). Unlike other members of this protein family, GPIHBP1 contains an N-terminal domain with 21 acidic amino acids (Glu, Asp). We propose that the first 30–35 N-terminal residues in GPIHBP1 have a very high propensity for being an intrinsically disordered region (*Figure 1B*). The disordered nature of the acidic domain is supported by: (1) the atypically large hydrodynamic volume of GPIHBP1$^{1–131}$ compared with GPIHBP1$^{34–131}$ (*Figure 1C*); (2) its

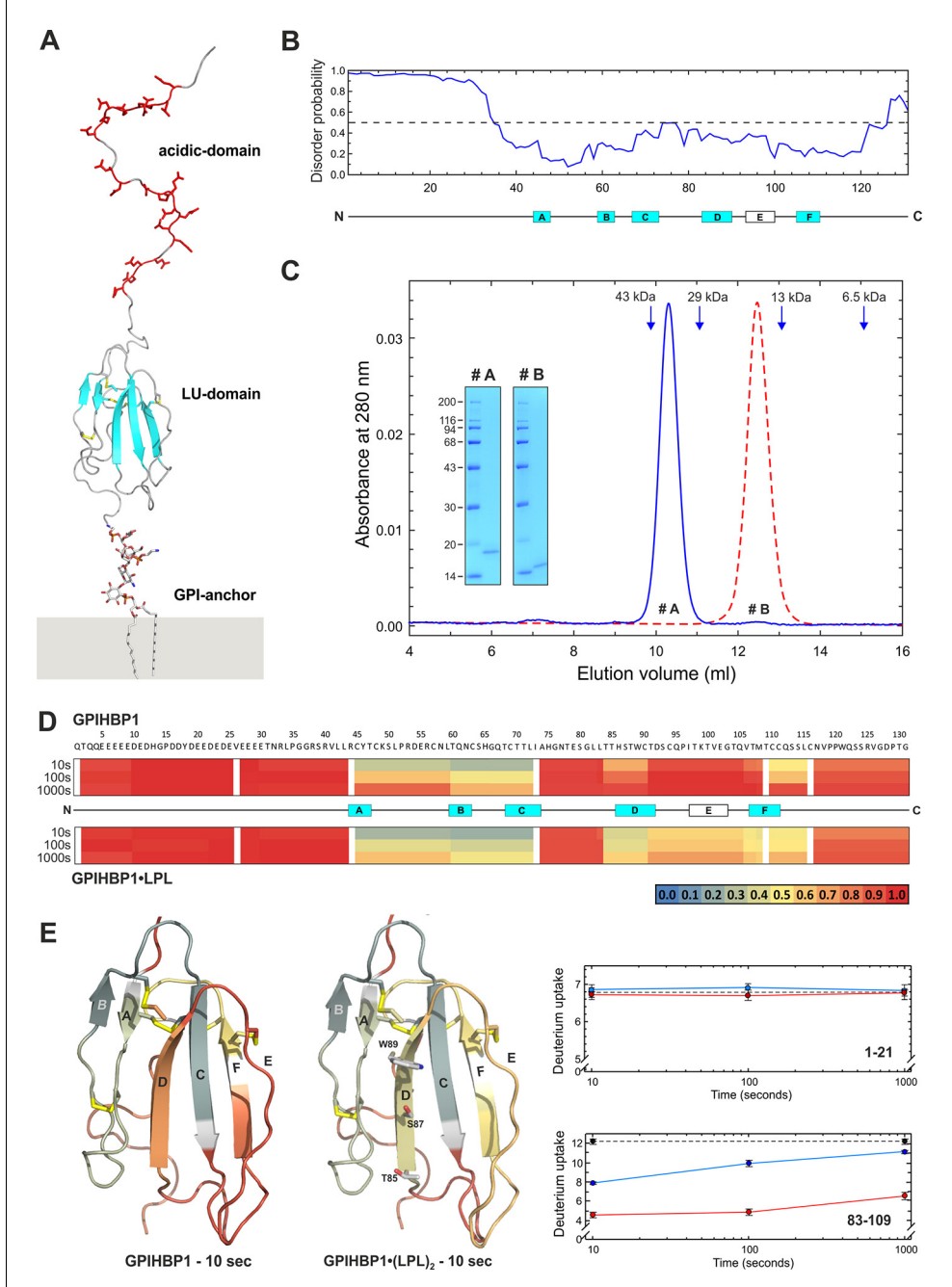

**Figure 1.** Model of human GPIHBP1. Panel **A** shows a cartoon representation for glycolipid-anchored human GPIHBP1. Predicted β-sheets are shown in cyan; acidic amino acid residues in the N-terminal domain are highlighted by red sticks; and the consensus disulfide bonds are shown in yellow. Panel **B** shows a "disorder prediction" for human GPIHBP1 sequence based on the IUPred algorithm. Locations of the six strands of the three-fingered–fold of the LU domain are highlighted by boxes (A–F); strands predicted to form β-sheets are colored cyan. Panel **C** documents the homogeneity and monomer status of purified GPIHBP1$^{1–131/R38G}$ (#A) and GPIHBP1$^{34–131/R38G}$ (#B) by analytical size-exclusion chromatography with a Superdex HR75 column operated with 20 mM NaH$_2$PO$_4$ and 150 mM NaCl (pH 7.2) and SDS-PAGE (*inset*). Elution positions of the calibration standards are indicated by blue arrows: ovalbumin (43 kDa), carbonic anhydrase (29 kDa), ribonuclease (13 kDa), and aprotenin (6.5 kDa). Panel **D** provides a heat map representation of the relative deuterium uptakes (relative to a fully exchanged control) in peptic peptides from free and LPL-occupied GPIHBP1$^{1–131}$ as assessed by HDX-MS. Deuterium uptake was measured after 10, 100, and 1000 s incubations in D$_2$O, and relative deuterium uptake is assigned according to the color code (ranging from *blue,* no deuterium uptake, to *red*, full deuterium uptake). A

*Figure 1 continued on next page*

*Figure 1 continued*

cartoon representation of the differential deuterium uptake for the LU domain between free and LPL-occupied GPIHBP1 after a 10-s exchange is shown in panel **E**. Shown as sticks are the positions of Thr[85], Ser[87], and Trp[89] in β-strand D of GPIHBP1. Raw deuterium uptake values for peptides 1–21 and 83–109 are shown for free and LPL-occupied GPIHBP1 (*blue* and *red* solid lines, respectively). The dashed line represents a "full deuteration" control. GPI, glycosylphosphatidylinositol; MS, hydrogen–deuterium exchange mass spectrometry; LU, Ly6/uPAR; SDS-PAGE, sodium dodecyl sulfate polyacrylamide gel electrophoresis

The following figure supplements are available for figure 1:

**Figure supplement 1.** Constructs encoding soluble DIII-ent-hGPIHBP1[1–131/R38G] and DIII-ent-hLPL[313–438].

**Figure supplement 2.** Purification of recombinant soluble human GPIHBP1[1–131/R38G] and GPIHBP1[34–131/R38G].

**Figure supplement 3.** Comparison of the dynamics in GPIHBP1[34–131] and GPIHBP1[1–131] in HDX-MS experiments.

**Figure supplement 4.** Peptide list for pepsin-treated GPIHBP1.

susceptibility to limited proteolysis after Arg[33] or Arg[38]; and (3) by the highly dynamic nature of the acidic domain as judged by extremely rapid hydrogen–deuterium exchange rates. In hydrogen–deuterium exchange/mass spectrometry (HDX-MS) studies, we observed 100% deuterium uptake in this domain even after the shortest exposure time (10 s) (*Figure 1D and 1E*), which is consistent with the predicted deuterium uptake for a disordered GPIHBP1[1–33] peptide (87% after 1 s and 100% after 10 s). In the same experiment, HDX profiles of peptides within the LU domain followed the secondary structure prediction. The deuterium uptake in the isolated LU domain in GPIHBP1[34–131] is indistinguishable from that of full-length GPIHBP1[1–131] (*Figure 1—figure supplement 3*), which implies that the acidic N-terminal region has little or no effect on the structure of GPIHBP1's LU domain.

## Characterizing the GPIHBP1•LPL interaction with HDX-MS

To identify protein–protein binding interfaces and/or uncover conformational changes associated with GPIHBP1•LPL binding, we determined the hydrogen–deuterium exchange profiles for GPIHBP1, LPL, and GPIHBP1•LPL complexes. The GPIHBP1•LPL complexes were formed by incubating 5 μM GPIHBP1 with 5 μM LPL homodimers (LPL$_2$) for 15 min in 10 mM Na$_2$HPO$_4$, 150 mM NaCl (pH 7.4) at 25°C before monitoring solvent exchange after dilution into D$_2$O for 10, 100, or 1000 s. We recovered 22 peptides from GPIHBP1 and 92 peptides from LPL after on-line pepsin digestion of quenched and tris (2-carboxyethyl) phosphine (TCEP)-reduced proteins at pH 2.5, which correspond to 100% and 87% sequence coverage, respectively (*Figure 1–figure supplement 4* and *Figure 2—figure supplement 1*).

As documented by the heat maps in *Figure 1D*, the conformation of the LU domain in GPIHBP1[1–131] was less dynamic when bound to LPL. In particular, we observed markedly reduced deuterium uptake within strands D and E of GPIHBP1 when it formed a complex with LPL (*Figure 1E*). This shift in the dynamics likely reflects a transition to a state with increased secondary structure and/or a direct engagement in a ligand-binding interface. The fact that the HDX-MS analysis of GPIHBP1 identified β-strand D as a candidate binding interface for LPL was not entirely unexpected given that a previous mutagenesis study had shown that individual alanine replacements of Thr[85], Ser[87], or Trp[89] in β-strand D impaired LPL binding (*Figure 1E*) (*Beigneux et al., 2011*). Moreover, two missense mutations involving β-strand D have been associated with chylomicronemia in humans (i.e. Ser[87]→Pro [*Buonuomo et al., 2015*] and Thr[88]→Arg [*Surendran et al., 2012*]). Mutating residues within strand E of GPIHBP1 had little impact on LPL binding (*Beigneux et al., 2011*). The reduced deuterium uptake that we recorded for strand E in the presence of LPL probably relates to a higher propensity for β-sheet formation in the setting of the GPIHBP1•LPL complex. Strand E is one of the most versatile structures in LU proteins; it can adopt a random coil (snake venom α-neurotoxins), a β-strand (uPAR domains I and II), or an α-helix (CD59, Prod1, and uPAR domain III) (*Kriegbaum et al., 2011*). Interestingly, the deuterium uptake in the intrinsically disordered N-terminal acidic domain did not change with LPL binding. That finding suggests that the acidic domain

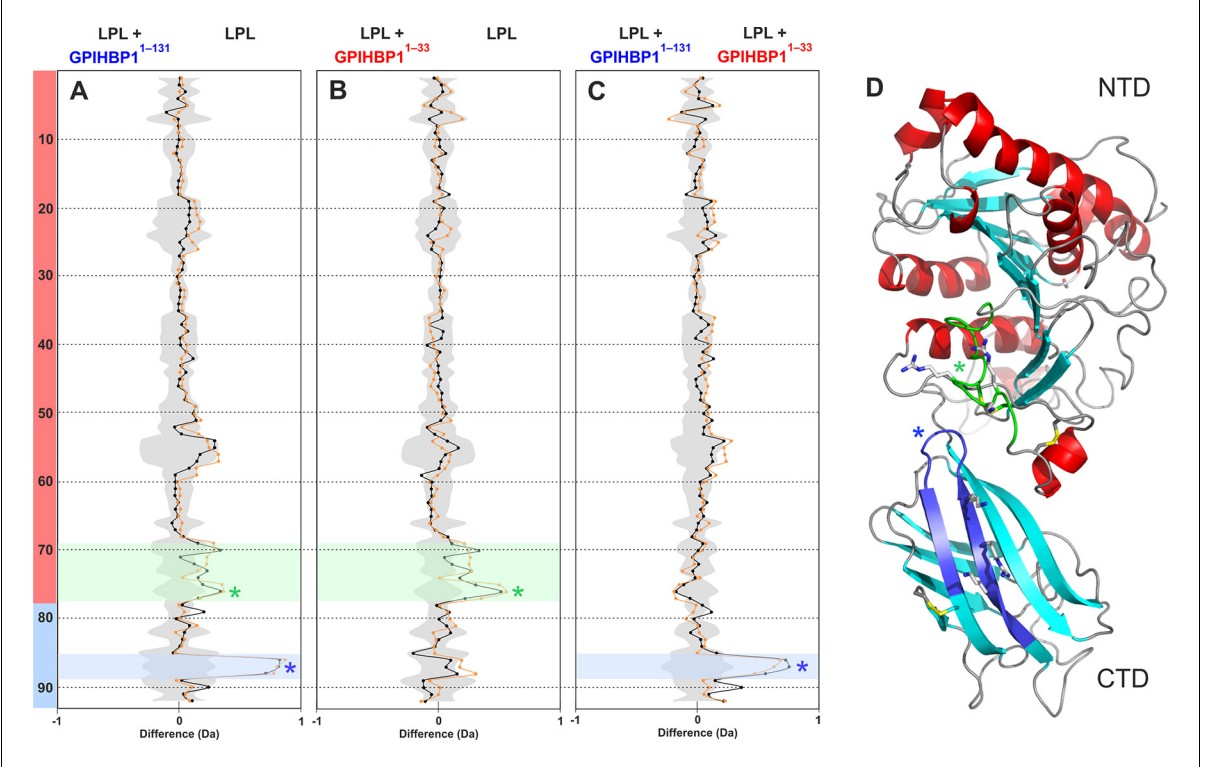

**Figure 2.** Mapping GPIHBP1 interaction sites on LPL. Differential deuterium uptake values for free LPL and LPL occupied with intact GPIHBP1$^{1-131}$ (panel **A**) or the acidic domain peptide GPIHBP1$^{1-33}$ (panel **B**) are shown as butterfly representations. The butterfly plot in panel **C** compares LPL occupied with GPIHBP1$^{1-131}$ or GPIHBP1$^{1-33}$. Due to inherent instability of LPL homodimers, the deuterium uptake was only examined at 10- (*orange*) and 100-s (*black*) incubations. The data points represent the mean of triplicate measurements, and the shaded gray area corresponds to the largest standard deviation in the data sets recorded for each peptide. A total of 92 peptides were recovered from LPL, and they are numbered consecutively from the N-terminus. The sequences of the individual peptides are found in *Figure 2—figure supplement 1B*. The transparent red and cyan colors on the left in panel **A** localize peptides to either the NTD or the CTD domains of LPL, respectively. Panel **D** shows a cartoon representation of human LPL. Two regions having the most pronounced changes in deuterium uptake with GPIHBP1 binding are highlighted in *green* (residues 279–293) and *blue* (residues 402–419); basic residues are shown as sticks. CTD, C-terminal domain; LPL, lipoprotein lipase; NTD, N-terminal domain.

The following figure supplement is available for figure 2:

**Figure supplement 1.** Peptide list for pepsin-treated bovine LPL.

either does not participate in the binding interaction or more likely is involved in very transient binding events with LPL (interactions that do not lead to the formation of stable hydrogen bonds).

From the same set of experiments, we also extracted data concerning the effects on LPL by comparing the differential deuterium uptake between unoccupied LPL and LPL in complex with either intact GPIHBP1$^{1-131}$ (*Figure 2A*) or its N-terminal acidic domain peptide GPIHBP1$^{1-33}$ (*Figure 2B*). To minimize possible confounding effects from the inherent instability of unoccupied LPL, we only included deuterium uptake values for 10 and 100 s for the three different states of LPL. Comparing these states, we identified residues 402–419 in the CTD of LPL as the most likely interaction site for the LU domain of GPIHBP1 (shown in blue in the butterfly plot in *Figure 2A*). This assignment is based on the fact that GPIHBP1$^{1-131}$, but not GPIHBP1$^{1-33}$, attenuates the deuterium uptake in this particular region (*Figure 2C*). Two missense mutations in or close to this segment (C418Y and E421K) were identified in patients with chylomicronemia (*Henderson et al., 1998*; *Henderson et al., 1996*), and follow-up studies showed that both mutations impaired GPIHBP1 binding (*Voss et al., 2011*). It is possible that additional regions in the CTD of LPL are involved in GPIHBP1 binding but escaped detection in our HDX-MS experiments due to the lower level of sequence coverage for the CTD (69%) (*Figure 2—figure supplement 1*). The impact of the acidic domain peptide (GPIHBP1$^{1-33}$) on LPL deuterium uptake was less pronounced than with intact GPIHBP1$^{1-131}$ (*Figure 2A and*

**Table 1.** Kinetic rate constants for GPIHBP1 interactions with CTD and LPL homodimers.[a]

| | $k_{on}$ $(10^5\ M^{-1}s^{-1})$ | $k_{off}$ $(s^{-1})$ | $K_D$ $(\mu M)$ | n | Capture |
|---|---|---|---|---|---|
| GPIHBP1[1–131/R38G] | 10.4 ± 1.9 | 0.12 ± 0.04 | 0.12 ± 0.04 | 11 | CTD |
| GPIHBP1[1–131] | 11.2 ± 4.0 | 0.22 ± 0.07 | 0.29 ± 0.13 | 10 | CTD |
| GPIHBP1[34–131/R38G] | 1.0 ± 0.6 | 0.10 ± 0.01 | 1.41 ± 0.76 | 4 | CTD |
| GPIHBP1[1–131/W89S] | nbd | nbd | nbd | 2 | CTD |
| GPIHBP1[1–131/R38G] N-gly[b] | 10.6 | 0.16 | 0.15 | 2 | CTD |
| | | | | | |
| GPIHBP1[1–131/R38G] | 9.9 ± 5.0 | 0.023 ± 0.006 | 0.025 ± 0.007 | 8 | LPL |
| GPIHBP1[1–131] | 6.5 ± 4.4 | 0.020 ± 0.007 | 0.038 ± 0.029 | 8 | LPL |
| GPIHBP1[34–131/R38G] | 1.9 ± 0.4 | 0.019 ± 0.005 | 0.091 ± 0.012 | 4 | LPL |
| | | | | | |
| GPIHBP1[1–131/R38G] | nbd | nbd | nbd | 1 | CTD[C418Y] |
| GPIHBP1[34–131/R38G] | nbd | nbd | nbd | 1 | CTD[C418Y] |

[a]The kinetic rate constants ($k_{on}$ and $k_{off}$) were derived by global fitting of sensorgrams obtained by either single-cycle kinetics (LPL) or a mixture of single- and multi-cycle kinetics (the CTD of LPL); the dissociation equilibrium constant $K_D$ was calculated as $k_{on}/k_{off}$.

[b]Intact GPIHBP1[1–131/R38G] (2 nmols) was incubated overnight at 37°C with 2 U N-glycanase under native conditions. The deglycosylated protein remained monomeric, as judged by analytical size-exclusion chromatography (as performed in **Figure 1C**).

n, number of separate determinations, each involving a complete set of analyte concentrations analyzed on different days and/or different CM4 chips; nbd, no specific binding detected; CTD, C-terminal domain; LPL, lipoprotein lipase.

**2B**), consistent with a more dynamic interaction (i.e. a shorter residence time of the peptide on LPL). Notwithstanding the more transient interaction, we did nevertheless observe that GPIHBP1[1–33] reduced deuterium uptake in LPL peptides spanning residues 279–293 (**Figure 2B**), which is localized in the domain interface between the NTD and CTD of LPL (**Figure 2D**). It is noteworthy that this region contains one of the heparin-binding sites in LPL enriched in basic residues (Arg[281], Lys[282], Arg[284]) (**Hata et al., 1993**) and therefore represents a potential binding partner for GPIHBP1's acidic domain.

## Kinetic rate constants for the interaction between GPIHBP1 and LPL

To dissect the individual contributions of the folded LU domain and the intrinsically disordered acidic domain on the binding kinetics with LPL, we developed and optimized a SPR assay to measure this interaction. In brief, we used a BiacoreT200 to measure the binding kinetics between soluble, monomeric GPIHBP1 and either LPL's CTD or intact LPL that had been captured on a CM4 sensor chip with an immobilized anti-LPL monoclonal antibody, 5D2. This antibody is well suited to capture and display both LPL and the CTD in a defined orientation because it recognizes both proteins with high affinity (**Chang et al., 1998**).

Using this SPR setup, we found that full-length GPIHBP1[1–131/R38G] binds the CTD of LPL with a $K_D$ of 0.12 ± 0.04 μM, whereas GPIHBP1[34–131/R38G] binds with a $K_D$ of 1.41 ± 0.76 μM (**Table 1**). The robustness of the assay is apparent by the repeat analyses of 250 nM GPIHBP1 at the end of each multi-cycle segment (blue curves in **Figure 3A and 3C**). In parallel, we analyzed the ability of soluble GPIHBP1 to bind to a CTD carrying a single amino acid substitution (C418Y). As noted earlier, the C418Y mutation is known to abolish LPL binding to GPIHBP1 (**Henderson et al., 1996**; **Voss et al., 2011**). When the LPL CTD[C418Y] was captured on immobilized mAb 5D2 (85 RU ~ 5 fmols/mm²), it lacked the ability to bind GPIHBP1[1–131] or GPIHBPB1[34–131], even at concentrations as high as 1 μM (**Table 1**). As an additional control, we tested the ability of GPIHBP1[1–131/W89S] to bind the CTD of LPL. Earlier studies had shown that glycolipid-anchored GPIHBP1[W89S] on the surface of transfected CHO-K1 cells has little or no capacity to bind LPL (**Beigneux et al., 2011**; **Beigneux et al., 2015**; **Plengpanich et al., 2014**). Consistent with those studies, the SPR measurements revealed negligible binding of GPIHBP1[1–131/W89S] to the CTD of LPL (**Figure 3B**). Collectively, these data indicate that

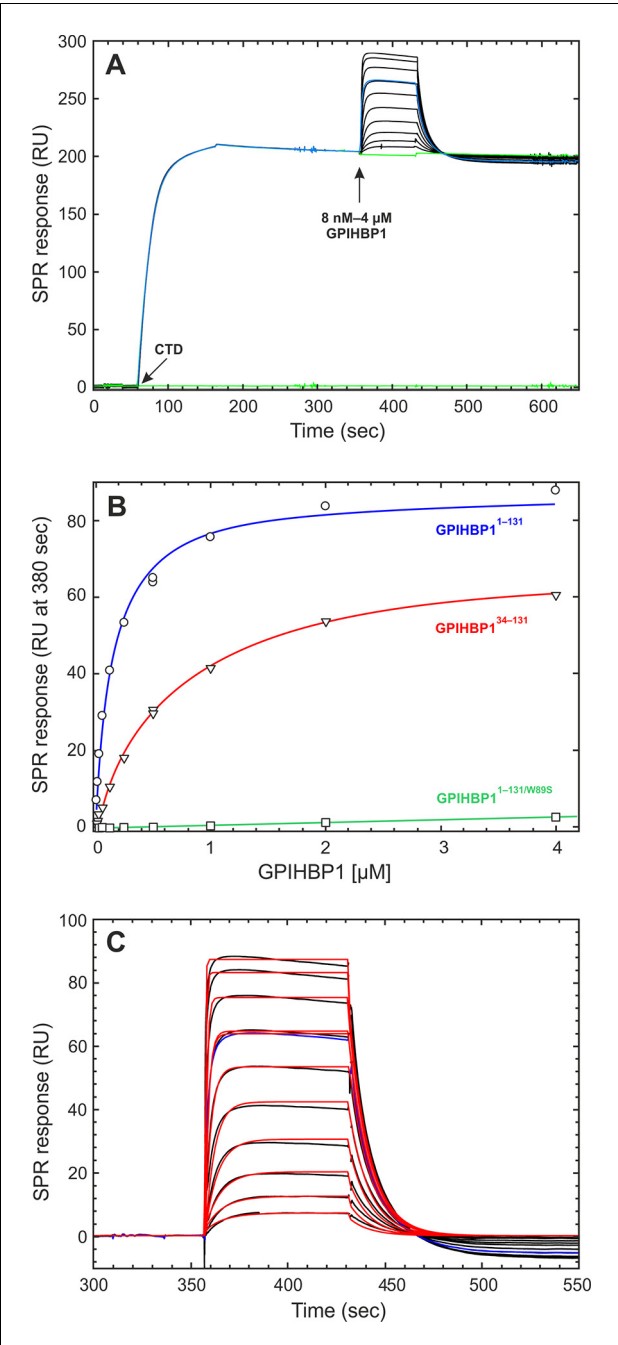

**Figure 3.** Real-time binding kinetics for the interaction between GPIHBP1 and the CTD of human LPL. Panel **A** shows repeat binding of 100 nM of recombinant CTD[313–448] from human LPL to immobilized mAb 5D2 followed by multi-cycle injections of twofold dilutions of GPIHBP1[1–131] (black lines). The green line represents a buffer control; the blue line represents a repeat injection of 250 nM GPIHBP1 at the end of the experiment to demonstrate reproducibility. Panel **B** shows equilibrium binding isotherms for the interactions between the immobilized CTD of LPL and GPIHBP1[1–131] (*blue*), GPIHBP1[34–131] (*red*), and GPIHBP1[1–131/W89S] (*green*) with the SPR signal at 380 s in *panel* A as equilibrium binding level. Panel **C** shows a kinetic evaluation of the double-referenced SPR data for GPIHBP1[1–131] with a global fit to a 1:1 binding model (fits shown in red). Note there is a slight decay of the binding signal at equilibrium due to a weak ligand-induced dissociation of the CTD from mAb 5D2; this effect translates into dissociation below baseline for the higher concentration of GPIHBP1[1–131]. These effects were not observed for GPIHBP1[34–131]. CTD, C-terminal domain; LPL, lipoprotein lipase; SPR, surface plasmon resonance.

the intrinsically disordered acidic domain of GPIHBP1 has little effect on the affinity of the LPL•GPIHBP1 interaction in the absence of a functional LU domain.

Kinetic assessment of the SPR binding data reveals that the 5–10-fold difference in the equilibrium binding constants between GPIHBP1$^{1–131}$ and GPIHBP1$^{34–131}$ could be accounted for by the differences in their association rate constants: $k_{on}$ = 1.04 × 10$^6$ M$^{-1}$s$^{-1}$ for GPIHBP1$^{1–131}$ versus a $k_{on}$ = 0.10 × 10$^6$ M$^{-1}$s$^{-1}$ for GPIHBP1$^{34–131}$ (Table 1). In contrast, the stabilities of the GPIHBP1•CTD complexes with GPIHBP1$^{1–131}$ and GPIHBP1$^{34–131}$ were essentially identical; both displayed $k_{off}$ values of 0.1 s$^{-1}$. Removal of the single N-linked glycosylation in GPIHBP1$^{1–131}$ had no significant impact on the kinetic rate constants (Table 1). As these analyses were conducted with the R38G mutation in GPIHBP1 (which was introduced to optimize purification yields), we tested whether this mutation affects the binding kinetics between LPL and GPIHBP1. To liberate intact, wild-type GPIHBP1$^{1–131}$, we performed a partial cleavage of 7 mg RS-DIII-ent-GPIHBP1 with 0.05 U enterokinase/mg protein, which resulted in 70% cleavage of the fusion protein. A sufficient amount of this full-length GPIHBP1$^{1–131}$ was purified by ion-exchange chromatography to allow subsequent kinetic analyses by SPR. The kinetics for the interactions between mAb 5D2-captured CTD or LPL and wild-type GPIHBP1$^{1–131}$ were similar to those measured with GPIHBP1$^{1–131/R38G}$ (Table 1). Thus, the R38G mutation does not significantly affect the properties of the GPIHBP1 interaction with LPL.

A similar kinetic assessment for the binding of LPL homodimers by multi-cycle titration was, however, inherently challenging because the conditions required to regenerate mAb 5D2 between cycles progressively reduced the capturing capacity of the sensor chip. To minimize this confounding factor, we performed a single-cycle kinetic titration of the interaction between GPIHBP1 and full-length LPL dimers captured by mAb 5D2. To be confident that the captured LPL was in a properly folded and homodimeric state, we initially performed a kinetic titration with two LPL-specific monoclonal antibodies (mAb 4-1a, which binds within residues 5–25 in LPL [Bensadoun et al., 2014] and mAb 5D2, which binds to residues 380–400 in the CTD of LPL [Chang et al., 1998]). As shown in Figure 4A, both antibodies bound immobilized LPL when analyzed by a single-cycle titration protocol. Calculation of their binding capacities [R$_{max}$ = 200 RU for mAb 4-1a (1.3 fmols/mm$^2$) and R$_{max}$ = 309 RU for mAb 5D2 (2.1 fmols/mm$^2$)] revealed molar surface-binding capacities that were similar to the surface density of the captured LPL homodimer (R$_{900sec}$ = 203 RU [2.0 fmols/mm$^2$]). Because mAb 5D2 is used as both capture and detection agent in this experiment, our data suggest that the majority of the captured LPL on the sensor chip was in the form of homodimers.

The single-cycle kinetic titrations revealed fast $k_{on}$ values for GPIHBP1•LPL interactions. The $k_{off}$ values for the LPL complexes with GPIHBP1$^{1–131}$ and GPIHBP1$^{34–131}$ were 5-fold slower than those for the corresponding CTD interaction, which indicates that GPIHBP1 forms a tighter complex with full-length LPL (Table 1, Figure 4B and 4C). To supplement the real-time binding kinetics recorded by SPR, we measured the equilibrium binding constants for the interaction between GPIHBP1$^{1–131/R38G}$ and LPL in solution by microscale thermophoresis (MST) with a Monolith NT.115 (NanoTemper Technologies GmbH; Germany). In these studies, 5 nM Alexa-647–labeled GPIHBP1$^{1–131/R38G}$ or GPIHBP1$^{34–131/R38G}$ were incubated with increasing concentrations of LPL$_2$ (10 pM to 350 nM). The binding isotherms were then calculated from the shifts in the thermophoresis of the fluorophore (Figure 5). With this experimental approach, we determined a K$_D$ of 5.7 nM for the LPL•GPIHBP1$^{1–131/R38G}$ interaction and a K$_D$ of 147 nM for the GPIHBP1$^{34–131/R38G}$ interaction. All things considered, we find that data with this solution-based assay aligns excellently with the binding affinities determined kinetically by SPR.

## Global unfolding of LPL's catalytic domain

We next used hydrogen–deuterium exchange to probe for conformational changes associated with the time-dependent decay of the catalytic activity in LPL (Osborne et al., 1985). For these studies, we incubated 5 µM LPL$_2$ at 25°C for various times and then traced changes in the solvent exposure of backbone amide hydrogens with a 10-s pulse labeling in D$_2$O. As illustrated in Figure 6, we observed a pronounced global unfolding of the catalytic domain in LPL as reflected by the appearance of a bimodal signature in the isotope envelopes from the majority of peptic peptides derived from this domain (colored red in Figure 6B). This finding implies that the NTD of LPL most likely enters a pre-molten globule-like state with little maintenance of secondary structure. The disordering of the catalytic triad is evident by the bimodal isotope envelope for the peptide 131–165, which

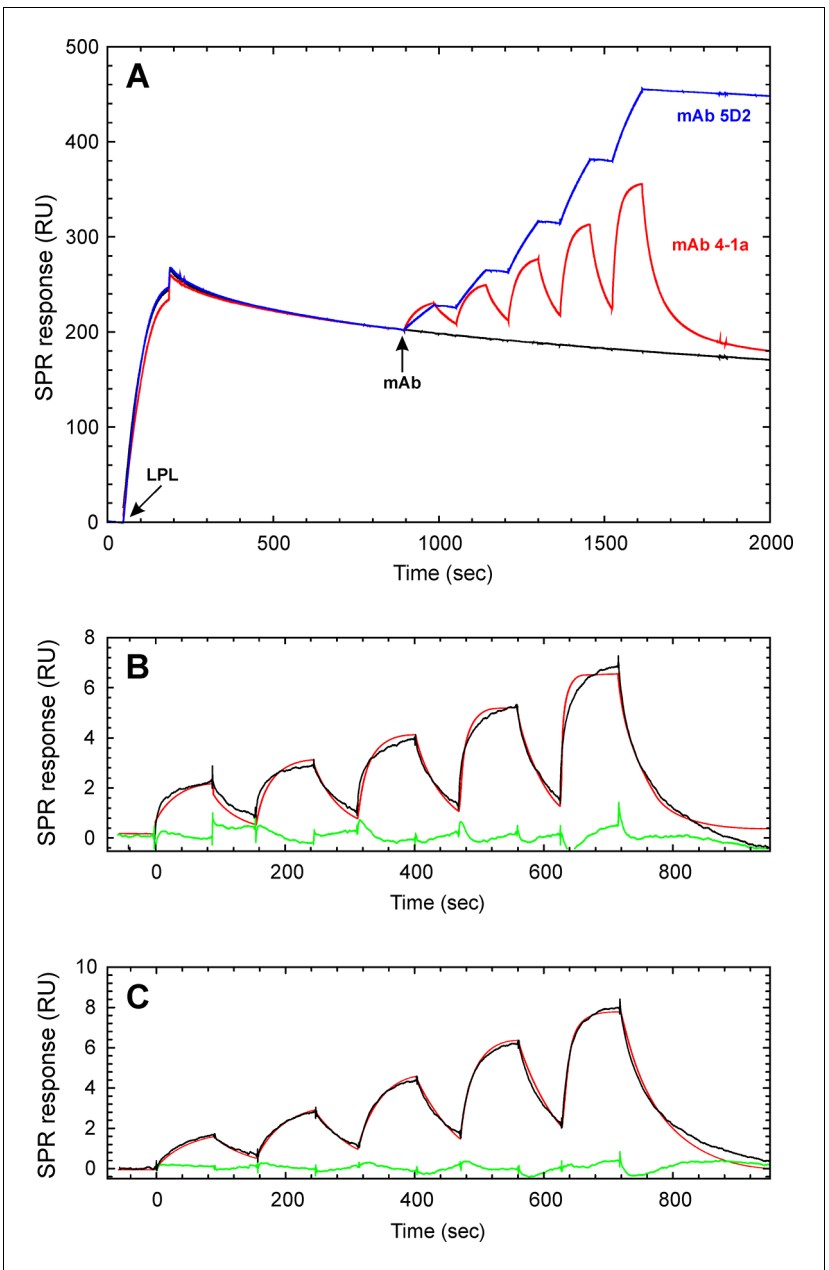

**Figure 4.** Kinetic assessment of the GPIHBP1•LPL interaction by single-cycle titration SPR. The basic principles in the single-cycle kinetic titration are illustrated in panel **A**. Initially, LPL is captured in a noncovalent fashion on the CM4 sensor surface after a 150 s injection of 200 nM of purified LPL across the flow cell containing immobilized mAb 5D2. After a 600 s stabilization period, a series of five 90 s pulses with increasing analyte concentration are injected without intervening regeneration. The following concentrations of either mAb 4-1a (**A**), mAb 5D2 (**A**), GPIHBP1$^{1-131}$ (**B**), or GPIHBP1$^{34-131}$ (**C**) were analyzed: 12.5, 25, 50, 100, and 200 nM. Panels **B** and **C** shows the buffer referenced sensorgrams recorded for GPIHBP1$^{1-131}$ and GPIHBP1$^{34-131}$, respectively. The mathematical fitting to a simple 1:1 binding model are superimposed as *red* lines and the residuals are shown in *green*. LPL, lipoprotein lipase; SPR, surface plasmon resonance.

encompasses active site residues Ser$^{134}$ and Asp$^{158}$ (*Figure 6A and 6B*). Of note, we also observed a time-dependent loss in both the triolein hydrolase and esterase activities of 2 μM LPL$_2$ with experimental conditions comparable to those for the HDX-MS studies (*Figure 6D*). Unexpectedly, the time course for the inactivation of LPL catalytic activity appeared to be approximately twice as fast as the

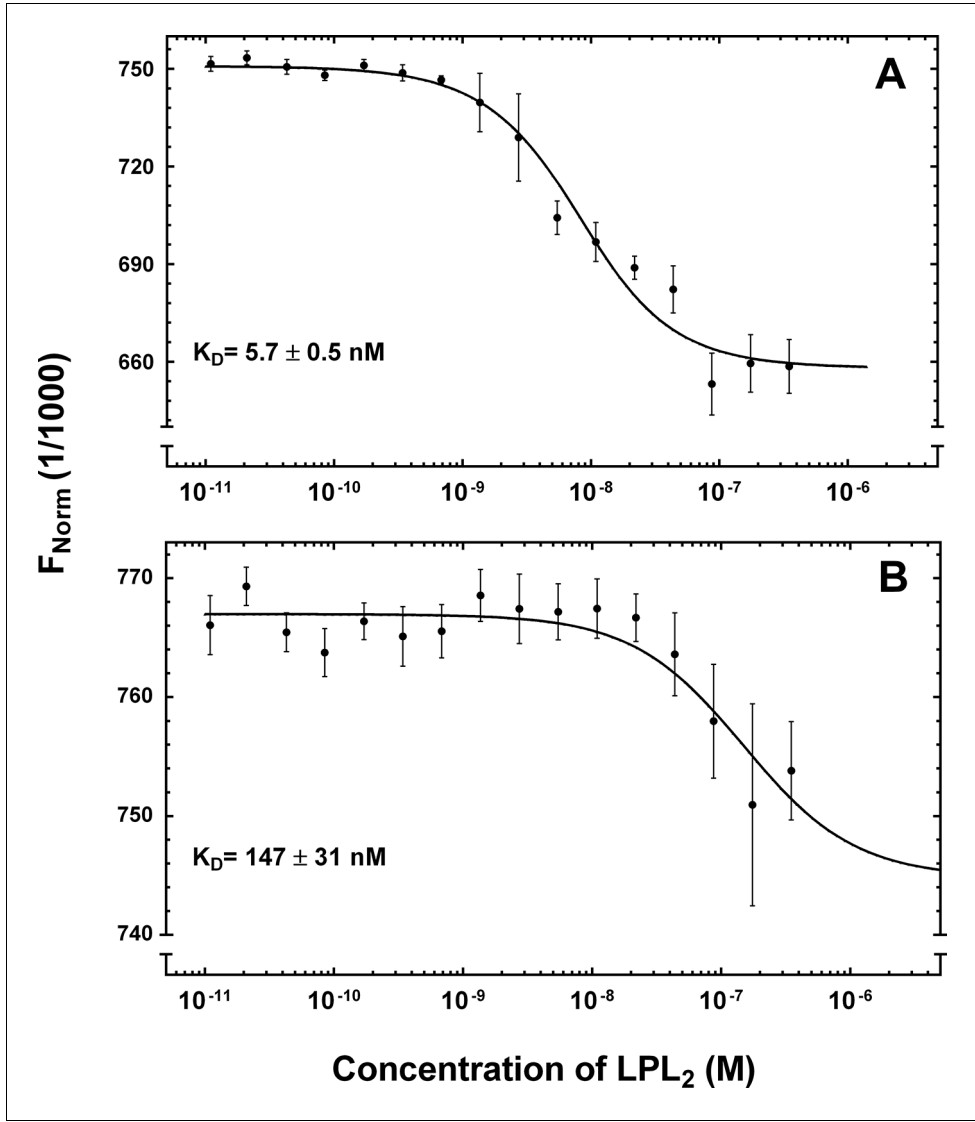

**Figure 5.** Equilibrium binding constants of the GPIHBP1•LPL interaction by microscale thermophoresis. The microscale thermophoresis signals for the interaction of LPL with 5 nM Alexa Flour-647–labeled GPIHBP1$^{1-131/R38G}$ (panel **A**) and GPIHBP1$^{34-131/R38G}$ (panel **B**) were recorded in quadruplicates for two-fold dilution series of unlabeled LPL$_2$ (10 pM to 350 nM). The mean values and standard deviation for the thermophoresis are shown as well as a fitting to a 1:1 binding model (software supplied with Monolith NT.115). LPL, lipoprotein lipase.

unfolding of the catalytic NTD documented by HDX-MS. With prolonged incubations, we did observe a more complete unfolding of the NTD by HDX-MS (74% unfolded after 180 min and 90% after 240 min). Of note, no changes were observed in the deuterium uptake of the CTD in LPL during the same time frame (*Figure 6B*), consistent with a greater stability of that domain (*Gin et al., 2012*).

## GPIHBP1$^{1-33}$ attenuates the rate of LPL unfolding

The impact of GPIHBP1 on the spontaneous unfolding of LPL was subsequently addressed by HDX-MS. This was accomplished by incubating LPL in the presence of GPIHBP1$^{1-131}$, GPIHBP1$^{1-33}$, or GPIHBP1$^{34-131}$ for 45 min at 25°C. We assessed the deuterium uptake after 10-s labeling with D$_2$O. GPIHBP1$^{1-131}$ clearly inhibited LPL unfolding (*Figure 6C*). This protective effect was predominantly due to the acidic domain because the GPIHBP1$^{1-33}$ peptide alone also attenuated LPL unfolding. A

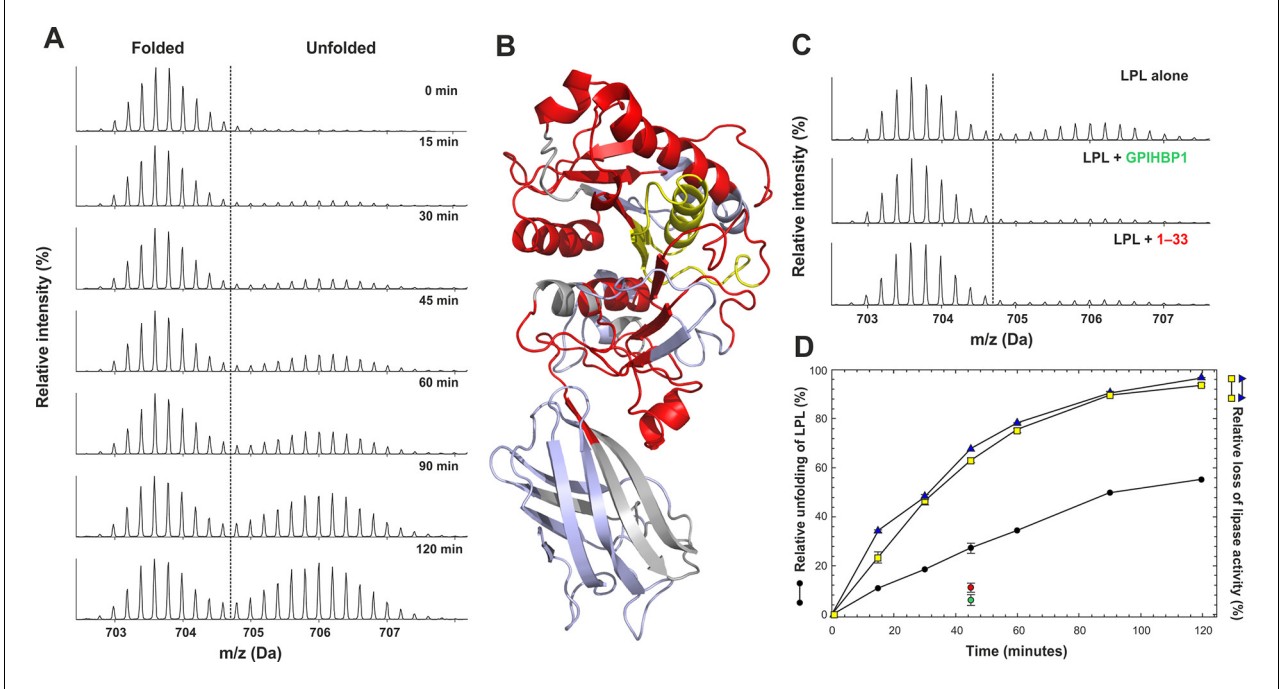

**Figure 6.** Progressive unfolding of LPL as determined by HDX-MS. Panel **A** shows the unfolding of the catalytic domain of LPL when incubated at 25°C. The unfolding is evident from the appearance of a bimodal isotopic envelope for the peptide 131–165, which contains Ser[134] and Asp[158] of the catalytic triad. Panel **B** shows the global distribution of peptides in the catalytic domain that undergo unfolding (in *red*). The peptide 131–165 is highlighted in *yellow*. Peptides that are not exhibiting bimodal isotope envelopes are colored *light blue*; segments of LPL not recovered by the HDX-MS are colored *gray*. The impact of GPIHBP1[1–131] and the N-terminal acidic peptide GPIHBP1[1–33] on LPL unfolding is shown in panel **C**. In these studies, equimolar amounts of GPIHBP1[1–131] or GPIHBP1[1–33] (relative to the LPL2) inhibited unfolding of the NTD of LPL. The progressive unfolding of LPL was quantified by fitting two Gaussian distributions to the isotopic envelopes of peptide 131–165 representing the folded and unfolded states (panel **A**) and is shown in panel **D** (*black circles*). Unfolding at the 45-min incubation time point was measured in triplicates with and without equimolar amounts of GPIHBP1[1–131] (*green circle*) or GPIHBP1[1–33] (*red circle*). The progressive loss of triolein hydrolase and esterase activities of LPL were recorded in parallel as a time-dependent function of pre-incubating 2 µM LPL2 under identical conditions and is shown by the *yellow squares* and *blue triangles*, respectively. LPL, lipoprotein lipase; HDX-MS, hydrogen–deuterium exchange mass spectrometry; NTD, N-terminal domain

distinct but much less pronounced contribution was provided by GPIHBP1[34–131] on both unfolding rates and preservation of triolein hydrolase and esterase activity of LPL (*Table 2*).

## Mapping the binding site for GPIHBP1[1–33] on LPL

To map defined binding sites for GPIHBP1 on LPL, we used zero-length covalent cross-linking with N-ethyl-N′-[3-diethylamino)propyl]-carbodiimide (EDC), which forms an isopeptide bond by chemical condensation of carboxylates and primary amino groups that are in close spatial proximity. For these studies, we formed GPIHBP1•LPL complexes by incubating a high concentration of LPL with a two-fold molar excess of GPIHBP1[1–131] or GPIHBP1[34–131] (to saturate binding sites on LPL). Adding 10 mM EDC efficiently cross-linked LPL•GPIHBP1[1–131] in a 1:1 complex, consuming the majority of the LPL (*Figure 7*, lanes 5 & 11). In contrast, no covalent adducts were formed in the samples containing GPIHBP1[1–131] alone, LPL alone, or a mixture of LPL and GPIHBP1[34–131] (*Figure 7*, lanes 3, 4, 6 & 10). Adding EDC to LPL in the presence of a 10-fold molar excess of GPIHBP1[1–33] also led to the formation of a 1:1 complex with no evidence of higher-order adducts (*Figure 7*, lane 9). These studies imply that each LPL molecule interacts with only a single acidic domain peptide.

To identify the specific cross-linked sites responsible for the formation of the covalent LPL•GPIHBP1 adduct, we performed an in-gel trypsin digestion of the complex that had been separated by sodium dodecyl sulfate polyacrylamide gel electrophoresis (SDS-PAGE) (*Figure 7*, lane 5). The extracted tryptic peptides were analyzed by a high-resolution Q-Exactive HF mass spectrometer, and cross-linked peptides were identified by their parent ion mass as well as subsequent HCD

**Table 2.** Preservation of LPL structure and activity by GPIHBP1 binding.

| Added ligand | Unfolding of NTD in LPL (%)[a] | Loss of triolein hydrolase activity (%)[b] | Loss of esterase activity (%)[b] |
|---|---|---|---|
| GPIHBP1[1–131/R38G] | 32.6 ± 5.7 | 28.7 ± 1.5 | 11.4 ± 6.7 |
| GPIHBP1[34–131/R38G] | 79.2 ± 4.2 | 51.5 ± 1.3 | 34.7 ± 8.4 |
| GPIHBP1[1–33] | 43.0 ± 5.1 | 4.2 ± 1.8[b] | 9.8 ± 6.9[b] |
| None | 100 | 100 | 100 |

[a]Inhibition of the spontaneous decay of LPL by GPIHBP1[1–131], GPIHBP1[34–131], or GPIHBP1[1–33] was measured after a 45-min incubation at 25°C. Equimolar amounts of GPIHBP1 and LPL homodimers were present during the incubation step.

[b]Triolein hydrolase activity was measured with a [³H]triolein substrate; esterase activity was measured with a soluble fluorescent substrate. Activity assays were performed on LPL samples that were subjected to the same pre-incubation conditions used for the HDX-MS experiments[a] except that *GPIHBP1[1–33]* was present in 5-fold molar excess compared to LPL. The protection by GPIHBP1 variants is shown relative to the decay of LPL after a 45-min incubation without any added binding partner (defined as 100%). Measurements were performed in triplicate.

fragment spectra (*Figure 7—figure supplement 1*). All but one of the five identified cross-links was established between LPL and GPIHBP1's acidic domain, consistent with the abundance of EDC-reactive carboxyl groups in the acidic domain. The corresponding cross-linking sites on LPL included Lys[298] in the catalytic domain and Lys[416], Lys[424], and Lys[430] in the CTD of LPL (*Figure 7B*). The one cross-linked peptide identified outside of the acidic domain occurred between residues 100–125 (most likely Glu[102]) in the LU-domain of GPIHBP1 and residues 423–430 of LPL. We did not detect cross-links between GPIHBP1[34–131] and LPL using 10 mM EDC for 90 min (*Figure 7*, lanes 6 & 10); however, we did observe such cross-links (*Figure 7—figure supplement 2*) if we drove the conjugation chemistry to higher yields by converting the *o*-acylisourea EDC intermediate to a more stable amine-reactive derivative (N-hydroxysuccinimide (NHS)-ester).

## Discussion

The interaction between LPL and GPIHBP1 is essential for lipolysis of TRLs along the capillary lumen (*Beigneux et al., 2007*; *Davies et al., 2010*; *Goulbourne et al., 2014*). In the absence of functional GPIHBP1, LPL is mislocalized within the interstitial spaces, leading to severe hypertriglyceridemia and reduced delivery of lipid nutrients to parenchymal cells (*Goulbourne et al., 2014*; *Young and Zechner, 2013*). *LPL* or *GPIHBP1* mutations that prevent LPL•GPIHBP1 binding or abolish LPL catalytic activity result in familial chylomicronemia in humans (*Buonuomo et al., 2015*; *Gin et al., 2012*; *Henderson et al., 1996*; *Plengpanich et al., 2014*; *Rios et al., 2012*; *Voss et al., 2011*). While the physiologic function of GPIHBP1 has gradually come into focus, our understanding of LPL•GPIHBP1 interactions was negligible. No three-dimensional structures had been determined for LPL or GPIHBP1, almost certainly because of the low stability of LPL and earlier difficulties in preparing pure recombinant GPIHBP1 free of protein dimers and multimers (*Beigneux et al., 2015*).

In the current study, we eliminated one of the roadblocks to progress by developing protocols for expressing and purifying human GPIHBP1. Our approach took advantage of *Drosophila* S2 cells as host cells for heterologous expression. Other LU domain–containing proteins have been expressed with this system (*Gårdsvoll et al., 2007*; *Gårdsvoll et al., 2004*), and the high quality of those protein preparations are highlighted by both the structures solved by X-ray crystallography (*Lin et al., 2010*; *Llinas et al., 2005*; *Xu et al., 2012*; *Zhao et al., 2015*) and the structure–function insights gained by biophysical approaches (HDX-MS, SAXS, and SPR) (*Gårdsvoll et al., 2006*; *Jørgensen et al., 2004*; *Mertens et al., 2012*). In the case of human GPIHBP1, we were able to prepare large quantities of highly purified, monomeric proteins for full-length GPIHBP1[1–131] as well as a truncated GPIHBP1 lacking the acidic domain (GPIHBP1[34–131]). The latter protein was generated by limited proteolysis of full-length GPIHBP1.

With the availability of homogeneous preparations of catalytically active LPL and pure, monomeric GPIHBP1, we uncovered several novel properties of the GPIHBP1•LPL interaction, all relevant to the physiology of intravascular TRL processing. We found: (1) that the N-terminal acidic domain of GPIHBP1 is intrinsically disordered; (2) that the acidic domain has a major effect on the association

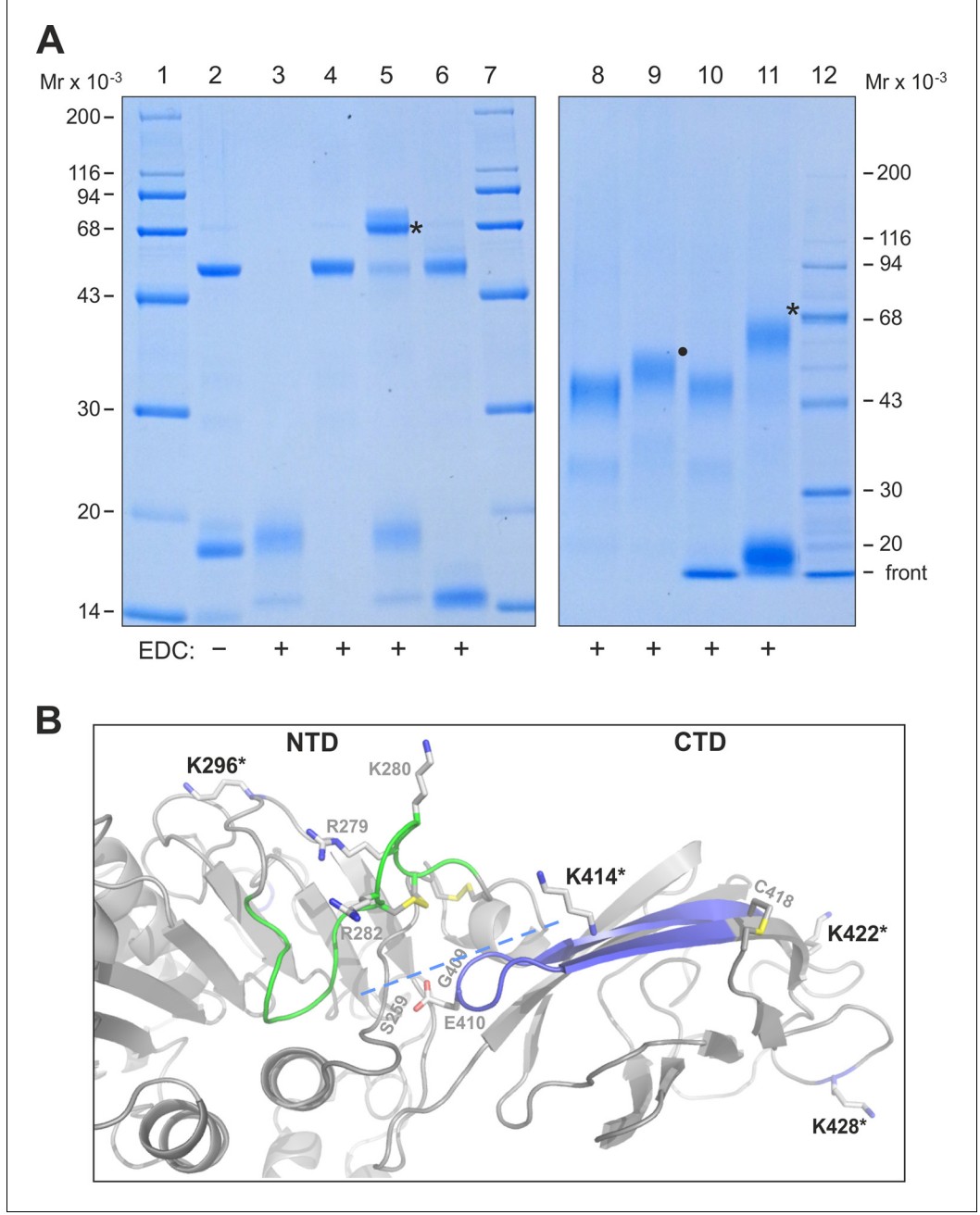

**Figure 7.** Zero-length cross-linking of GPIHBP1$^{1–131}$ and GPIHBP1$^{1–33}$ to bovine LPL. Panel **A**, *left* shows a Coomassie Blue–stained 12% polyacrylamide gel after SDS-PAGE analysis of reduced and alkylated samples representing various combinations of 1.5 µM LPL$_2$ and 7 µM GPIHBP1 variants subjected to EDC cross-linking. *Lane 2* shows LPL GPIHBP1$^{1–131}$ before cross-linking. Lanes 3–6 show samples after EDC cross-linking: GPIHBP1$^{1–131}$ (*lane 3*); LPL (*lane 4*), LPL GPIHBP1$^{1–131}$ (*lane 5*), and LPL GPIHBP1$^{34–131}$ (*lane 6*). The covalently bound conjugate representing LPL•GPIHBP1$^{1–131}$ is marked by an asterisk. *Right panel* shows a Coomassie Blue–stained 4–12% gradient polyacrylamide gel after analysis of EDC cross-linked 3 µM LPL$_2$ alone (*lane 8*) or in the presence of 15 µM GPIHBP1$^{1–33}$ (*lane 9*); 15 µM GPIHBP1$^{34–131}$ (*lane 10*); and 15 µM GPIHBP1$^{1–131}$ (*lane 11*). The covalent conjugates representing LPL•GPIHBP1$^{1–131}$ and LPL•GPIHBP1$^{1–33}$ are indicated by an asterisk and a solid dot, respectively. Molecular weight markers are shown in *lanes 1, 7 & 12*. Panel **B** shows a model of human LPL highlighting the cross-linking sites in GPIHBP1 that were identified by MS (*asterisks*). Areas that have been assigned as potential interaction sites for GPIHBP1 by HDX-MS are shown in *green* (for the acidic domain of GPIHBP1) and *blue* (for the LU domain of GPIHBP1). The position of the interdomain interface in LPL between the NTD and CTD is marked by a dashed line, and three residues within this interface linked to familial chylomicronemia when mutated (S259R, G409R, and E410V) are shown by gray numbers. Basic residues of the heparin-binding site in the catalytic domain of LPL (R279, K280, R282) are shown as sticks. Note, bovine LPL contains two additional residues compared with human LPL, for example Lys$^{296}$ in human LPL is equivalent to Lys$^{298}$ in bovine LPL. CTD, C-terminal domain; EDC, N-ethyl-N′-[3-diethylamino)propyl]-

*Figure 7 continued on next page*

*Figure 7 continued*

carbodiimide; HDX-MS, hydrogen–deuterium exchange mass spectrometry; LPL, lipoprotein lipase; MS, mass spectrometry; NTD, N-terminal domain; SDS-PAGE, sodium dodecyl sulfate polyacrylamide gel electrophoresis.

The following figure supplements are available for figure 7:

**Figure supplement 1.** Fragment spectra identifying EDC-mediated inter-domain cross-links in mature LPL•GPIHBP1$^{1–131}$ complexes.

**Figure supplement 2.** SDS-PAGE analysis of EDC/NHS cross-linked LPL•GPIHBP1 complexes.

rate constant for the GPIHBP1•LPL interaction; (3) that the time-dependent decay of LPL activity is associated with a global unfolding of LPL's catalytic domain; (4) that the acidic domain of GPIHBP1 preserves the catalytic activity of LPL by mitigating global unfolding of the catalytic domain; and (5) that two discrete interaction sites in the LPL•GPIHBP1 complex cooperate to promote ligand binding and stabilization of LPL catalytic activity.

Our findings have allowed us to conceptualize a model for unoccupied GPIHBP1 in which the N-terminal acidic domain is intrinsically disordered and the remainder of the protein (i.e. GPIHBP1$^{34–131}$) adopts the prototypical three-finger-fold characteristic of LU domain proteins (*Figure 1A*). This model was initially suggested by the observation that full-length GPIHBP1 (GPIHBP1$^{1–131}$) exhibits an unexpectedly large hydrodynamic volume (as judged by its elution profile in size-exclusion chromatography [*Figure 1C*]), whereas a truncated GPIHBP1 lacking the acidic domain (GPIHBP1$^{34–131}$) elutes at the expected position for a folded globular protein. The atypical elution profile for the full-length GPIHBP1 is indicative of a large Stokes radius and is one of the hallmarks of intrinsically disordered proteins (*Uversky, 2012*). Further evidence for the disordered nature of the acidic domain is provided by its highly dynamic state as measured by HDX-MS. Our model offers potential insights into the interplay between GPIHBP1 and LPL during LPL mobilization and TRL processing in vivo. In GPIHBP1-deficient mice, the LPL in tissues is mislocalized to the interstitial spaces and never reaches the luminal surface of capillary endothelial cells. It is noteworthy that the LPL in GPIHBP1-deficient mice remains sequestered in the interstitial space, presumably bound to negatively-charged heparin sulfate proteoglycans (HSPGs) (*Davies et al., 2010*). This interaction is thought to be driven by electrostatic interactions and is characterized by fast kinetic rate constants (*Lookene et al., 1996*), presumably creating a dynamic reservoir of LPL within the interstitial spaces. In wild-type mice, newly secreted LPL likely binds to the same HSPGs but then moves to GPIHBP1 on capillary endothelial cells. We propose that the intrinsically disordered and markedly acidic N-terminal extension of GPIHBP1 plays an important role in mobilizing HSPG-bound LPL within the interstitial spaces through long-ranged electrostatic interactions, thereby driving LPL into association with endothelial cells. This model for LPL mobilization by GPIHBP1 is consistent with the kinetic rate constants that we determined for GPIHBP1•LPL interactions, which are 10-fold faster for intact GPIHBP1$^{1–131}$ than for GPIHBP1$^{34–131}$ lacking the acidic domain.

Since the difference in $k_{on}$ for GPIHBP1$^{1–131}$ and GPIHBP1$^{34–131}$ is observed for both intact LPL and LPL's CTD alone, we propose that an electrostatic 'encounter complex' is initially established between GPIHBP1's acidic domain and the basic residues in LPL's CTD. This encounter complex then guides the maturation of the interaction, facilitating a tighter binding interface between GPIHBP1's LU domain and residues 402–419 in LPL (*Figure 2*). The cross-linking that we documented between residues 1–33 of intact GPIHBP1$^{1–131}$ and Lys$^{416}$, Lys$^{424}$, and Lys$^{430}$ in the CTD of LPL is consistent with this initial encounter/maturation model. The electrostatic component of this model is in accordance with a previous study (*Reimund et al., 2015*).

The existence of a dual binding mechanism is also consistent with the available mutagenesis data. First, the importance of the positively charged heparin-binding sequences in LPL's CTD has been demonstrated by reduced binding of hLPL$^{K403A;R405A;K407A;K413A;K414A}$ to GPIHBP1 (*Gin et al., 2008*). Second, there is ample evidence for the involvement of additional CTD sequences—apart from positively charged residues—in the binding of LPL to the LU-domain in GPIHBP1. For example, two missense mutations in LPL (C418Y and E421K), first identified in patients with chylomicronemia, have no effect on heparin binding but abolish binding to GPIHBP1 (*Henderson et al., 1998*; *Henderson et al., 1996*; *Voss et al., 2011*). Lending support to this two-step binding mechanism is

the fact that the acidic domain plays little or no role in the stability of established GPIHBP1•LPL complexes. The $k_{off}$ values for GPIHBP1$^{1-131}$ and GPIHBP1$^{34-131}$ are comparable, regardless of whether the binding is to LPL's CTD ($k_{off}$ = 0.1 s$^{-1}$) or to full-length LPL ($k_{off}$ = 0.02 s$^{-1}$).

A different two-site interaction model between GPIHBP1 and LPL was recently proposed (*Reimund et al., 2015*), which suggested that a single molecule of GPIHBP1 binds two LPL homodimers. This conclusion was based on SPR studies in which GPIHBP1 (from the medium of transfected CHO cells) had been captured onto chips coated with the anti-GPIHBP1–specific mAb 11A12. To obtain reliable fits for the LPL-binding profiles recorded by SPR and reduce nonspecific binding of LPL, these authors performed their analyses at 4°C and in the presence of 0.4–0.6 M NaCl. In the current study, the SPR data were obtained under more physiologic conditions (20°C in 0.15 M NaCl), and the binding profiles fit well to a simple 1:1 interaction model (i.e. GPIHBP1+LPL$_2$⇌ GPIHBP1•LPL$_2$). A likely explanation for the discrepancy between the two studies is the different designs of the SPR studies. The current study measured the binding of purified, monomeric GPIHBP1 in solution to purified LPL that had been captured by mAb 5D2 thereby avoiding the inherent difficulties in having LPL as a soluble analyte. The earlier study measured the interaction between LPL in solution to mouse GPIHBP1 (from the medium of transfected CHO cells) that had been captured on mAb 11A12. We now know the GPIHBP1 secreted from transfected CHO-cells contains significant amounts of dimers and multimers (*Beigneux et al., 2015*), a factor that would further complicate the interpretation of these SPR data.

An intriguing finding in the current study was the observation that LPL undergoes a time-dependent global unfolding of its catalytic N-terminal domain, as judged by the loss of stable secondary structure in HDX-MS studies (*Figure 7*). This transition into a pre-molten globule–like conformation of LPL's catalytic domain provides a plausible structural explanation for the time-dependent decline in LPL catalytic activity at room temperature (*Osborne et al., 1985*). It is noteworthy, however, that unfolding of the catalytic domain in LPL proceeds with only one-half the rate of the parallel loss of catalytic activity. That observation is consistent with a model where the unfolding of one LPL molecule in the homodimer would lead to the inactivation of both subunits.

From our HDX-MS analyses of GPIHBP1•LPL complexes (*Figure 2*), it is apparent that only one region in LPL responds to the presence of the acidic domain GPIHBP1$^{1-33}$, and that is located at the interface between the CTD and the catalytic domain (represented by the peptide spanning residues 279–290) (*Figure 7B*). The involvement of this 'interdomain interface' of LPL in binding GPIHBP1$^{1-33}$ was bolstered by zero-length cross–linking studies. The interaction between LPL Lys$^{298}$ (in close spatial proximity to peptide 279–290) and GPIHBP1's acidic domain was covalently trapped by EDC (*Figure 7B*). The fact that the acidic domain interacts with the domain interface of LPL in the setting of a mature GPIHBP1•LPL complex likely carries functional implications. Indeed, the stabilization of the catalytic domain of LPL by GPIHBP1$^{1-131}$ was primarily mediated by GPIHBP1$^{1-33}$ (*Table 1*). We suspect that the stabilization of LPL's 'interdomain interface' by GPIHBP1's acidic domain serves to limit protein dissociation/unfolding of the partner monomers of an LPL homodimer. Genetic evidence provides circumstantial support for the importance of a properly assembled interdomain interface for LPL activity. Certain homozygous missense mutations that are associated with reduced LPL activity and chylomicronemia [LPL$^{E410V}$, LPL$^{G409R}$, and LPL$^{S259R}$ in humans; LPL$^{G412R}$ in cats) (*Foubert et al., 1997*; *Ginzinger et al., 1996*; *Kassner et al., 2015*; *Previato et al., 1994*)] are located at the interdomain interface in close spatial proximity to the proposed binding site in LPL$^{279-290}$ for GPIHBP1$^{1-33}$ (*Figure 7B*). All mutations alter the electrostatics of the interface. In future studies, it will also be very interesting to investigate if the accelerated inactivation of LPL by ANGPTL4 (*Sukonina et al., 2006*) involves this interdomain region and the extent to which GPIHBP1 protects LPL from ANGTPL4-mediated inactivation.

From a practical point of view, our discovery that the acidic domain stabilizes the conformation of LPL by mitigating unfolding could prove to be helpful in future efforts to define the structure of LPL by X-ray crystallography. We suspect that incubating LPL with peptides derived from GPIHBP1's acidic domain could promote stabilization of LPL, increase its conformational homogeneity, and enhance the likelihood of growing well-diffracting crystals for X-ray structure determination.

## Materials and methods

### Purified proteins and reagents

Bovine LPL was purified from fresh bovine milk by heparin-Sepharose, hydroxyapatite, and Superdex HR200 size-exclusion chromatography as described (*Cheng et al., 1985*). Fractions containing dimeric LPL (LPL$_2$) were precipitated with 3.6 M NH$_4$SO$_4$ and the pellet dissolved in 5 mM phosphate buffer (pH 6.5) containing 40% (v/v) glycerol. Small aliquots were stored at –80°C until use. Recombinant enterokinase (1 U/μl) produced in *Pichia pastoris* (EKMax) was purchased from Invitrogen (Carlsbad, CA). Monoclonal antibodies against human uPAR (R2 and R24) and human LPL (5D2 and 4-1A) were produced and characterized as described (*Bensadoun et al., 2014*; *Chang et al., 1998*; *Gårdsvoll et al., 2011*). A synthetic 33-mer peptide (GPIHBP1$^{1–33}$) representing the N-terminal acidic domain of GPIHBP1 (QTQQEEEEEDEDHGPDDYDEEEDEVEEEETNR) was obtained at a purity of >95% from TAG-Copenhagen A/S (Copenhagen, Denmark).

### Expression and purification of human GPIHBP1 and LPL$^{313–348}$ by *Drosophila* S2–cells

A truncated version of human GPIHBP1 (residues 1–131), lacking the C-terminal signal peptide that normally triggers the covalent addition of a glycolipid anchor, was expressed by (and secreted from) *Drosophila* S2 cells as a fusion protein with uPAR domain III (RS-DIII-ent-GPIHBP1, *Figure 1—figure supplement 1A*). This strategy has been used for expressing other LU-domain proteins (*Gårdsvoll et al., 2007*; *Gårdsvoll et al., 2013*; *Hansen et al., 2004*). The N-terminal uPAR DIII serves the dual role of both the detection and the purification tag, and the DDDDK motif in the linker region allows excision of GPIHBP1 by enterokinase cleavage while preserving the original N-terminus of GPIHBP1. Using this strategy, we also expressed the mutants RS-DIII-ent-GPIHBP1$^{W89S}$ and RS-DIII$^{R281G}$-ent-GPIHBP1$^{R38G}$. The latter construct was designed to improve the overall yield of purified intact GPIHBP1 after enterokinase cleavage.

A similar platform was employed to express the CTD of human LPL (residues 313–448) using constructs RS-DIII-ent-CTD and RS-DIII-ent-CTD$^{C418Y}$ (*Figure 1—figure supplement 1B*).

After 7 days of induction with 0.5 mM CuSO$_4$, the medium from the *Drosophila* S2 cells was harvested and supplemented with 0.1 volume of 1 M Tris/HCl (pH 8.0) containing 0.2 M ethylenediaminetetraacetic acid (EDTA) and 0.1% (w/v) NaN$_3$ and 0.005 volume of 200 mM PMSF in DMSO. The various RS-DIII-ent-GPIHBP1 proteins were purified from the medium of transfected cells by immunoaffinity chromatography with an anti-uPAR mAb R2 column with 0.1 M CH$_3$COOH, 0.5 M NaCl (pH 2.5) as the eluent. The purified proteins were exchanged into 20 mM NaH$_2$PO$_4$, 150 mM NaCl (pH 7.2) and stored at –80°C. The RS-DIII-ent-CTD constructs were affinity purified with an anti-uPAR mAb R24 column (*Gårdsvoll et al., 2011*) and eluted with 0.1 M CH$_3$COOH, 0.5 M NaCl (pH 2.5) containing 20% (v/v) glycerol and then exchanged into 20 mM NaH$_2$PO$_4$, 150 mM NaCl (pH 7.2) containing 20% (v/v) glycerol to prevent precipitation. Average yields of the fusion proteins were 2–5 mg/l.

### Enterokinase cleavage and purification of human GPIHBP1

The DIII-tag was removed from the purified fusion proteins by adding 0.1 U enterokinase/mg and incubating at 37°C. An equal amount of enterokinase was added 6 hr later, and the incubation was continued for a total of 24 hr. This digest was dialyzed against 50 mM CH$_3$COOH (pH 4.5). The excised GPIHBP1 was purified by cation-exchange chromatography with a 5-ml HiTrap SP FF (GE Healthcare; Uppsala, Sweden) and a 35-ml linear NaCl gradient (from 0 to 1.0 M) in 50 mM CH$_3$COOH (pH 4.5) (*Figure 1—figure supplement 2A*). Relevant fractions were pooled and buffer-exchanged into 20 mM NaH$_2$PO$_4$, 150 mM NaCl (pH 7.2) before conducting a final purification step by size-exclusion chromatography on a Superdex HR75 column in the same buffer (*Figure 1C*).

### Affinity of the GPIHBP1•LPL interaction as assessed by surface plasmon resonance

To minimize confounding effects resulting from the dextran surface of the sensor chip on the real-time kinetics recorded for the LPL•GPIHBP1 interaction, we used a CM4 sensor chip with a low degree of carboxylation to immobilize the LPL-specific monoclonal antibody 5D2 (*Chang et al.,*

*1998*). As a further precaution, we used 1 M ethylenediamine rather than ethanolamine to block excess active NHS esters, which reduces the charge effects from the remaining unmodified carboxylates on the matrix (*Glaser et al., 2014*). Covalent immobilization of mAb 5D2 was accomplished by injecting 5 µg/ml 5D2 dissolved in 10 mM sodium acetate (pH 5.0) over a CM4 chip that had been pre-activated with NHS/EDC (N-ethyl-N´-[3-diethylamino)propyl]-carbodiimide), aiming at a surface density of 1500 resonance units (RU) corresponding to 10 fmols/mm$^2$.

After a 100-s loading pulse with 100 nM recombinant LPL CTD$^{313-448}$ at 20 µl/min, real-time interactions between 5D2-captured CTDs (∼ 100–200 RU; 6–12 fmol/mm$^2$) and serial twofold dilutions of purified human GPIHBP1 were measured from 8 nM to 4 µM at 20°C using 10 mM HEPES, 150 mM NaCl, 3 mM EDTA (pH 7.4) containing 0.05% (v/v) surfactant P20 as running buffer. To be confident of a high reproducibility, at least one GPIHBP1 concentration was re-tested at the end of each experiment. Between cycles, mAb 5D2 was regenerated with two consecutive 10-µl injections of 0.1 M acetic acid/HCl (pH 2.5) in 0.5 M NaCl and 20 mM H$_3$PO$_4$. The immobilized mAb 5D2 tolerated repetitive regenerations without significant decline of its capturing capacity.

After double-buffer referencing of the recorded real-time interaction analyses, the kinetic rate constants ($k_{on}$ and $k_{off}$) were derived by global non-linear regression fitting of the association and dissociation phases to a simple bimolecular interaction model, including correction for mass transport limitations assuming pseudo–first-order reaction conditions with BIA evaluation 4.1 software (Biacore, Uppsala, Sweden). In some cases, the equilibrium binding constants $K_D$ and $R_{max}$ were also calculated separately from the corresponding equilibrium binding isotherms by nonlinear curve fitting assuming saturation of a single binding site:

$$R_{eq} = (R_{max}[\text{GPIHBP1}])/(K_d + [\text{GPIHBP1}]),$$

where $R_{eq}$ is the binding level at equilibrium, and $R_{max}$ is the binding capacity of the chip.

To enable efficient capturing of intact dimeric bovine LPL (LPL$_2$) on immobilized mAb 5D2, the purified LPL$_2$ was diluted 500–1000 fold to 200 nM in 10 mM HEPES (pH 7.4), 150 mM NaCl, 4 mM CaCl$_2$, 0.05% (v/v) surfactant P20 supplemented with 0.2 mg/ml bovine serum albumin (BSA) and 1% (w/v) carboxymethyl dextran to stabilize LPL and avoid non-specific adsorption to the microfluidics during loading. The running buffer was equivalent to the loading buffer, except that no carboxymethyl dextran was added. Regeneration of the capturing mAb was accomplished by two consecutive injections of 10 µl of 20 mM glycine/HCl (pH 2.5) and 10 µl of 5% (v/v) HCOOH-containing 0.5 M NaCl. To minimize deterioration from repetitive regenerations, single-cycle kinetic titration (*Karlsson et al., 2006*) of LPL•GPIHBP1 interactions were recorded during five consecutive injections of 20 µl of two-fold dilutions of purified GPIHBP1 at 20°C. After double-buffer referencing of the data, the corresponding kinetic rate constants were derived by fitting the data to a simple bimolecular interaction model using the mathematic model developed for single-cycle kinetics (T200 Evaluation Software 2.0, GE Healthcare Life Science, Uppsala, Sweden).

## Equilibrium binding constants of the GPIHBP1•LPL interaction by microscale thermophoresis

To determine solution equilibrium binding constants between LPL and Alexa Flour-647–labeled versions of GPIHBP1$^{1-131/R38G}$ and GPIHBP1$^{34-131/R38G}$, we used the MST technology (*Jerabek-Willemsen et al., 2014*). Purified GPIHBP1$^{1-131/R38G}$ and GPIHBP1$^{34-131/R38G}$ preparations were labeled for 30 min at 37°C with Alexa Flour-647 NHS ester (Thermo Fisher Scientific) at a molar ratio of 1:3 in phosphate-buffered saline (pH 7.4); these conditions favor modification of the N-terminal α-amino group. The reaction was terminated by adding 10 mM ethanolamine, and the proteins were desalted on a PD-10 column (GE Healthcare, Uppsala, Sweden). The average degree of labeling was 1.3 flurophore/protein for both GPIHBP1$^{1-131/R38G}$ and GPIHBP1$^{34-131/R38G}$; both labeled proteins remained monomeric as judged by analytical size-exclusion chromatography on a 5/150 Superdex 75 column (GE Healthcare, Uppsala, Sweden). The equilibrium binding between 5 nM Alexa Flour-647–labeled GPIHBP1 and LPL were calculated from the change in thermophoresis ($\Delta F_{norm} = F_{Hot}/F_{cold}$) measured with a Monolith NT.115 (NanoTemper Technologies GmbH) after adding increasing concentrations of non-fluorescent LPL. A two-fold dilution series ranging from 10 pM to 350 nM LPL$_2$ was prepared in 10 mM HEPES (pH 7.4), 150 mM NaCl, 4 mM CaCl$_2$, 0.05% (v/v) surfactant P20, and 1.0 mg/ml BSA. Samples were loaded into low-binding hydrophilic capillary tubes and the

thermophoresis signals were measured at 22°C with a light-emitting diode (LED) power of 80% and an infrared (IR) laser power of 100%.

## Determination of lipase and esterase activity of purified LPL

Time-dependent inactivation profiles for LPL were determined by incubating 2 µM purified $LPL_2$ alone or in the presence of 2 µM $GPIHBP1^{1-131}$, 2 µM $GPIHBP1^{34-131}$, or 10 µM $GPIHBP1^{1-33}$ for various times at 25°C in 10 mM $Na_2HPO_4$, 150 mM NaCl (pH 7.4). The temperature-induced unfolding was quenched by adding ice-cold 100 mM Tris (pH 7.8) containing 5 mM deoxycholic acid (DOC) and 0.1 mM sodium dodecyl sulfate (SDS) (DOC/SDS buffer). Esterase activities were analyzed by mixing 5 µl of LPL diluted 100-fold in DOC/SDS buffer with 95 µl DGGR assay buffer to a final concentration of 50 mM Tris, 50 µM 1,2-o-dilauryl-rac-glycero glutaric acid-(6'-methylresorufin) ester (DGGR), 120 mM NaCl, 10 mg/ml BSA, 0.5% Triton X-100 (v/v), pH 7.4. Ester hydrolysis was determined by measuring the linear increase of resorufin fluorescence at $\lambda_{ex}$ 530 nm, $\lambda_{em}$ 590 nm during the initial 5 min. Lipase activities were determined by adding 5 µl of LPL diluted 1000-fold in DOC/SDS to 195 µl of incubation mixtures with Intralipid containing [$^3$H] triolein (*Larsson et al., 2013*).

## Protein dynamics measured by HDX-MS

All hydrogen–deuterium exchange reactions were performed at 25°C and 300 RPM mixing, using 10 mM $Na_2HPO_4$, 150 mM NaCl buffers in either $H_2O$ or $D_2O$, adjusted to pH 7.4 and pD 7.4 ($pH_{read}$ = 7.0), respectively. The following protein solutions were made with the $H_2O$ buffer: 5 µM $LPL_2$, 5 µM $GPIHBP1^{1-131/R38G}$, 5 µM $LPL_2$ 5 µM $GPIHBP1^{1-131/R38G}$, 5 µM $LPL_2$ 25 µM $GPIHBP1^{1-33}$. Most solutions were pre-incubated for 15 min before deuterium labeling to promote efficient complex formation. Samples containing exclusively $LPL_2$ were not pre-incubated because of concerns about the inherent instability of LPL homodimers.

Isotopic labeling was initiated by adding $D_2O$ buffer to the protein solutions to a final concentration of 70% $D_2O$ (vol/vol). After 10-, 100- or 1,000-sec, aliquots were collected from the labeling solutions. These were mixed with 1 volume ice-cold quenching buffer (100 mM $Na_2HPO_4$, 0.8 M TCEP, 2 M urea in $H_2O$, pH 2.5). Quenched samples were placed in an ice bath for 2 min to reduce disulfide bonds and subsequently snap-frozen in liquid nitrogen.

Full deuteration controls were prepared by (1) diluting LPL and GPIHBP1 to 10 µM in 10 mM $Na_2HPO_4$, 150 mM NaCl, 2 M urea, 70% (vol/vol) $D_2O$, pD 7.4; (2) incubating samples for 48 hr at 37°C; and (3) quenching as described above without urea to achieve an identical solvent composition in the quenched samples. Labeling was performed in triplicate for each sample combination.

## Unfolding of LPL assessed by pulse-labeling HDX-MS

Buffers and labeling conditions were identical to those described previously. The following protein solutions were prepared in $H_2O$ buffer: 5 µM $LPL_2$, 5 µM $LPL_2$ 5 µM $GPIHBP1^{1-131/R38G}$, 5 µM $LPL_2$ 5 µM $GPIHBP1^{1-33}$, and 5 µM $LPL_2$ 5 µM $GPIHBP1^{34-131/R38G}$. Solutions were incubated for 15, 30, 45, 60, 90, and 120 min, diluted to 70% $D_2O$ buffer, allowed to exchange for 10 s, quenched by adding 1:1 ice-cold quench buffer, incubated for 2 min in an ice bath, and then snap frozen. Labeling was performed in triplicate for each sample combination incubated at 45 min.

## MS of HDX-labeled samples and data analysis

Quenched and reduced samples were analyzed with a nanoACQUITY UPLC reversed-phased chromatographic system equipped with HDX technology (Waters, Milford, MA) coupled to a Synapt G2 electrospray ionization mass spectrometer (Waters). Desalting was performed by applying a flow of 300 µL/min buffer A (0.23% [v/v] formic acid [FA]) to an ACQUITY UPLC BEH C18 1.7-µm, 2.1 × 5 mm Vanguard Pre-Column by an Agilent 1260 Infinity Quaternary pump (Agilent Technologies, Santa Clara, CA). Peptides were separated on a 1.0 × 100 mm ACQUITY UPLC BEH C18 1.7-µm analytical column by a 12-min gradient from 95% buffer A to 50% buffer B (0.23% [v/v] FA in acetonitrile) at a flow of 40 µl/min. Proteins were digested online with an Upchurch guard column (1.0 × 20 mm, IDEX, Oak Harbor, WA) packed with agarose-immobilized pepsin (Thermo Scientific Pierce, Rockford, IL). Peptides from peptic digests were identified from DDA MS/MS runs using ProteinLynx Global Server v2.4 (Waters) and MassAI v1.07 (MassAI Bioinformatics, Stenstrup, DK, http://www.

massai.dk). Deuterium incorporation for intact proteins and peptides was quantified with DynamX V2.0 (Waters).

## Homology modeling and disorder prediction for GPIHBP1

Human GPIHBP1 (UniProt id: Q8VI16) without the N- and C-terminal signal sequences was used for a homology search with the program HHPred (*Soding, 2005*). To find suitable reference structures, the PDB database was used to generate a multiple sequence alignment with human GPIHBP1; the top five highest-ranking proteins (PDB entries: 20l3, 2h7z, 2h5f, 3neq, 1hcp) were selected as templates for homology modeling with MODELLER (*Sali et al., 1995*). Molecular structures are displayed by PyMOL (Schrödinger, LLC). A search for regions in human GPIHBP1 with a high propensity for intrinsic disorder was performed with the disorder prediction tools IUPred (*Dosztanyi et al., 2005*) and DISOPRED3 (*Jones and Cozzetto, 2015*).

## Zero-length chemical cross-linking of GPIHBP1•LPL complexes

To initiate covalent cross-linking, mixtures of 1.5 µM $LPL_2$ with either 7 µM $GPIHBP1^{1–131}$, 7 µM $GPIHBP1^{34–131}$, or 15 µM $GPIHBP1^{1–33}$ were exposed to 10 mM EDC for 90 min at 25°C in 10 mM $Na_2HPO_4$ (pH 8.0), 150 mM NaCl, 0.1% (v/v) Triton X-100, and 10% (v/v) glycerol. The cross-linking reaction was terminated by boiling in SDS-PAGE sample buffer; samples were subsequently reduced and alkylated before analysis by SDS-PAGE to assess the formation of covalent complexes. Cross-linked tryptic peptides were identified by high-resolution mass spectrometry with a Q-Exactive HF mass spectrometer (Thermo Scientific).

## In-gel digestion and desalting of cross-linked LPL•GPIHBP1$^{1–131}$ complexes

Coomassie Blue–stained protein bands corresponding to cross-linked protein complexes were excised from the polyacrylamide gel and subjected to in-gel reduction, alkylation and digestion with trypsin, as described with minor modifications (*Shevchenko et al., 1996*). Gel pieces were cut into smaller pieces, washed twice in 50% ethanol (v/v) and shrunk with 96% (v/v) ethanol. The shrunken gel pieces were then swollen in a 10 mM dithiothreitol (DTT) 0.1 M $NH_4HCO_3$ solution and incubated for 45 min at 56°C. After reduction, the solution was cooled to room temperature. The excess DTT solution was removed and replaced with a 55 mM iodoacetamide (IAA) 0.1 M $NH_4HCO_3$ solution followed by incubation for 30 min in the dark at room temperature. Excess IAA solution was removed; the gel pieces were washed twice in 50% ethanol (v/v) and the gel pieces were shrunken again using 96% (v/v) ethanol. Finally, the gel pieces were rehydrated on ice with a 100 mM $NH_4HCO_3$ solution containing 12.5 ng/µl trypsin. After 45 min, excess trypsin solution was removed and replaced with 100 mM $NH_4HCO_3$, and the gel pieces were then incubated at 37°C overnight. The supernatant containing tryptic peptides was recovered the next day. Remaining peptides in the gel were extracted by adding 5% (v/v) formic acid, incubating for 15 min, and then adding an equal volume of acetonitrile and incubating for an additional 15 min. This supernatant was combined with the original supernatant and the entire pool dried in a vacuum centrifuge. Prior to analysis, the tryptic digest was resuspended in 0.1% trifluoroacetic acid (TFA) and desalted using solid-phase extraction in a pipet tip packed with Oligo R3 reversed phase resin (Applied Biosystems). The column was subsequently washed with 0.1% (v/v) TFA; the bound sample was eluted with 0.1% (v/v) TFA in 70% (v/v) acetonitrile and then dried in a vacuum centrifuge.

## Reversed-phase nano-LC-ESI-MS/MS

The samples were resuspended in 0.1% (v/v) FA and loaded onto an EASY-nLC system (Thermo Scientific) with a two-column setup. Trapping was performed with a 100 µm × 2 cm Acclaim Pep-Map100 C18 Nano-Trap Column and peptides were separated along a 75 µm × 25 cm Acclaim PepMap RSLC analytical column (both Thermo Scientific). The peptides were eluted with an organic solvent gradient from 95% phase A (0.1% [v/v] FA) to 25% phase B (0.1% [v/v] FA, 95% [v/v] acetonitrile) over 80 min, followed by a 10-min gradient to 50% phase B at a flow rate of 250 nl/min. The eluted peptides were analyzed with a Q-Exactive HF mass spectrometer (Thermo Scientific), operated in the positive ionization mode with data-dependent acquisition. Each MS scan was acquired at

a resolution of 60,000 FWHM followed by 20 high-resolution HCD-MS/MS scans of the most intense ions. Ions selected for MS/MS were dynamically excluded for durations of 10 sec.

## Data analysis

MGF files were generated from the raw data with Proteome Discoverer v1.3 (Thermo Scientific), and the protein crosslinks identified using MassAI (*Peng et al., 2014*) (www.MassAI.dk). Searches against a database containing bovine LPL and GPIHBP1 R38G were performed with the following parameters: Precursor mass tolerance 10 ppm, MS/MS mass tolerance of 0.05 Da, tryptic cleavage allowing up to two missed cleavage sites, and carbamido-methylation of cysteine as a fixed modification. The following variable modifications were allowed: oxidation of Met; phosphorylation of Ser, Thr, and Tyr; and N-glycosylation of Asn-X-Ser/Thr/Cys. Peptide cross-links between Lys and Asp/Glu, with a resulting loss of water, were allowed. Cross-links were considered to be true if they had a score above 20 and the MS/MS spectra either contained fragment ion series originating from both peptides or contained peaks that represented cross-linked fragment ions.

## Acknowledgements

The coordinates for the model of human LPL were provided by Dr. Z. Liu (School of Pharmaceutical Sciences, Beijing, China). We acknowledge Gry Rasmussen, Haldis Egholm, and Gitte Juhl Funch for expert technical assistance and John Post for artwork. This work was supported by a Leducq Transatlantic Network grant (12CVD04) and by NIH grants HL090553 and HL087228. The authors have no financial interests to declare.

## Additional information

### Competing interests

SGY: Reviewing editor, *eLife.* The other authors declare that no competing interests exist.

### Funding

| Funder | Grant reference number | Author |
|---|---|---|
| Leducq Transatlantic Network | Grant 12CVD04 | Anne P Beigneux<br>Fong G Loren<br>Stephen G Young<br>Michael Ploug |
| National Institutes of Health | HL090553 | Stephen G Young |
| National Institutes of Health | HL087228 | Stephen G Young |

The funders had no role in study design, data collection and interpretation, or the decision to submit the work for publication.

### Author contributions

SM, Designed, performed, and analyzed the HDX-MS and Orbitrap-MS experiments with input from TJDJ and MP; KKK, Performed experiments related to modeling, prediction, SPR, and EDC crosslinking; ML, Performed LPL activity assays; APB, HG, Developed expression protocols for recombinant proteins; LGF, AB, Supplied essential reagents; TJDJ, Assisted in the design and intepretation of HDX-MS data; SGY, Conceived and wrote the manuscript along with MP with input from all co-authors.; MP, Conceived and supervised the study and wrote the manuscript along with SGY with input from all co-authors.

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
