## [Decision Letter]

Thank you for submitting your work entitled "The acidic domain of GPIHBP1 stabilizes lipoprotein lipase activity by preventing unfolding of its catalytic domain" for consideration by *eLife*. Your article has been favorably evaluated by Harry Dietz (Senior editor) and three reviewers, one of whom is a member of our Board of Reviewing Editors.

Tobias Walter (peer reviewer) has agreed to reveal his identity; the other two reviewers remain anonymous.

The reviewers have discussed the reviews with one another and the Reviewing Editor has drafted this decision to help you prepare a revised submission.

The manuscript from Ploug et al. investigates structural properties and interactions of purified GPIHBP1 and LPL using hydrogen-deuterium exchange, surface plasmon resonance, zero length crosslinking and enzyme activity assays. The authors use results of these studies in conjunction with homology modeling to propose a two-domain model for GPIHBP1 and its interaction with LPL. The model is further informed by studies of mutant proteins associated with hyperchylomicronemia. The authors propose that GPIHBP1 is comprised of a disordered acidic N terminal domain connected to a prototypical LU domain that mediates glycolipid anchoring to the plasma membrane. Both domains of GPIHBP1 are found to influence interactions with LPL, with the N-terminal interaction domain suggested to stabilize the catalytic domain of LPL and reduce loss of enzymatic activity upon warming to 20 degrees C. The authors interpret their findings with respect to the requirement of GPIHBP1 to move LPL from its site of secretion to the capillary lumen where it carries out lipolysis of triglyceride rich lipoproteins.

All three reviewers found the manuscript to be an important contribution to the understanding of triglyceride-rich lipoprotein metabolism. One reviewer commented, 'The investigation is a tour-de-force in uncovering the structural aspects of the interaction of GPIHBP1 with lipoprotein lipase.' A second reviewer commented, 'Besides highlighting the physiological role of this protein pair, this study is interesting for revealing a biochemical mechanism that coulFn1d be used to regulate lipases'.

While the overall impression was that the studies were performed at a high level of rigor and that the conclusions appear solid, two concerns were raised that would need to be satisfactorily addressed for acceptance.

1) Kinetics of the GPIHBP1-LPL interaction rely exclusively on SPR measurements; these measurements can be affected by the surface binding of ligands and the specific orientation of the protein. Thus, it would be important to independently assay binding constants in a solution assay. For steady state measurements, ITC could work; otherwise, the new knowledge of binding sites could be used for FRET measurements etc.

2) This is somewhat less important, but much of the paper relies on the analysis of a GPIHBP1 mutant. The authors point out that there are organisms where the mutated residue is found in the wild type protein, arguing this residue change is neutral. However, there are other changes in the homologue protein and it remains formally possible (although unlikely) that the mutant version of the protein does not behave the same way as the wild type. Maybe there is enough of the wt protein that can be expressed for competition experiments (with the mutant)? Alternatively can the authors show expressing the wt and mutant protein in CHO cells (as done before) that both alleles encode proteins with similar binding affinities. For instance, one could mix cells with WT and mutant GPIHBP (labeled with a fluorophore), incubate with LPL and assess binding in the same reaction.

---

## [Author Response]

*1) Kinetics of the GPIHBP1-LPL interaction rely exclusively on SPR measurements; these measurements can be affected by the surface binding of ligands and the specific orientation of the protein. Thus, it would be important to independently assay binding constants in a solution assay. For steady state measurements, ITC could work; otherwise, the new knowledge of binding sites could be used for FRET measurements etc.*

We agree that recording high-content, real-time binding kinetics with surface plasmon resonance is a challenging task, particularly if one of the binding partners is a hydrophobic, unstable, heparin-binding protein such as LPL. For this reason, we spent considerable time and effort (before performing any measurements) to optimize the SPR assay conditions so as to avoid confounding effects from the SPR matrix. Nevertheless, we agree with the reviewers that an additional approach involving equilibrium binding kinetics would be useful to validate/substantiate our SPR data. One technology that we previously considered for such studies was microscale thermophoresis (MST). MST is a solution-based micro-method that allows a high degree of freedom in choosing the optimal buffer conditions. In new experiments, we labeled GPIHBP1 with 1.3 Alexa-647 fluorophores per GPIHBP1 molecule. We used a neutral pH during labeling chemistry to favor the modification of the α-amino group of the N-terminal acidic domain, and we made sure that the labeled protein remained monomeric. Using the MST approach, we measured a K_D_ of 5.7 nM for GPIHBP^1–131/R38G^ and 147 nM for GPIHBP1^34–131/R38G^ for equilibrium LPL–GPIHBP1 binding. These data are entirely consistent with the findings and the conclusions drawn from the SPR analyses—although the absolute values differ slightly. In the revised manuscript, we have described the new data in the Results section (subsection “Kinetic rate constants for the interaction between GPIHBP1 and LPL”, last paragraph), added a new Figure (Figure 5), and expanded the Methods section accordingly (subsection “Equilibrium binding constants of the GPIHBP1•LPL interaction by microscale thermophoresis”).

*2) This is somewhat less important, but much of the paper relies on the analysis of a GPIHBP1 mutant. The authors point out that there are organisms where the mutated residue is found in the wild type protein, arguing this residue change is neutral. However, there are other changes in the homologue protein and it remains formally possible (although unlikely) that the mutant version of the protein does not behave the same way as the wild type. Maybe there is enough of the wt protein that can be expressed for competition experiments (with the mutant)? Alternatively can the authors show expressing the wt and mutant protein in CHO cells (as done before) that both alleles encode proteins with similar binding affinities. For instance, one could mix cells with WT and mutant GPIHBP (labeled with a fluorophore), incubate with LPL and assess binding in the same reaction.*

We agree that introducing a mutation to obtain a better yield of full-length GPIHBP1 could represent a possible confounding factor (even though homology considerations suggested that the risk was low). We have therefore produced the D3-ent-GPIHBP1 with the wild-type sequence and liberated GPIHBP1 with very low amounts of enterokinase (0.05 U/mg). The full-length wild-type GPIHBP^1–131^ was then purified by cation-exchange chromatography and used for SPR studies of GPIHBP1 binding to full-length LPL and LPL’s carboxyl-terminal domain. These new data recapitulate those recorded for the R38G mutant; thus, the R38G substitution has no significant impact on GPIHBP1–LPL binding kinetics. The new data have been added to Table 1 and are discussed in the Results section (subsection “Kinetic rate constants for the interaction between GPIHBP1 and LPL”, third paragraph).